# UNC-43/CaMKII-triggered anterograde signals recruit GABA$_A$Rs to mediate inhibitory synaptic transmission and plasticity at *C. elegans* NMJs

Yue Hao [1,2,3,6], Haowen Liu[4,6], Xian-Ting Zeng[1], Ya Wang[5], Wan-Xin Zeng[1,2,3], Kang-Ying Qian[1,2,3], Lei Li[4], Ming-Xuan Chi[1], Shangbang Gao [5], Zhitao Hu [4] & Xia-Jing Tong [1] ✉

Disturbed inhibitory synaptic transmission has functional impacts on neurodevelopmental and psychiatric disorders. An essential mechanism for modulating inhibitory synaptic transmission is alteration of the postsynaptic abundance of GABA$_A$Rs, which are stabilized by postsynaptic scaffold proteins and recruited by presynaptic signals. However, how GABAergic neurons trigger signals to transsynaptically recruit GABA$_A$Rs remains elusive. Here, we show that UNC-43/CaMKII functions at GABAergic neurons to recruit GABA$_A$Rs and modulate inhibitory synaptic transmission at *C. elegans* neuromuscular junctions. We demonstrate that UNC-43 promotes presynaptic MADD-4B/Punctin secretion and NRX-1α/Neurexin surface delivery. Together, MADD-4B and NRX-1α recruit postsynaptic NLG-1/Neuroligin and stabilize GABA$_A$Rs. Further, the excitation of GABAergic neurons potentiates the recruitment of NLG-1-stabilized-GABA$_A$Rs, which depends on UNC-43, MADD-4B, and NRX-1. These data all support that UNC-43 triggers MADD-4B and NRX-1α, which act as anterograde signals to recruit postsynaptic GABA$_A$Rs. Thus, our findings elucidate a mechanism for pre- and postsynaptic communication and inhibitory synaptic transmission and plasticity.

Fast synaptic inhibition is mediated by neurotransmitter GABA and GABA-activated chloride channels (GABA$_A$Rs). Disturbed inhibitory synaptic transmission has functional impacts on the pathology of neurodevelopmental and psychiatric disorders (including autism spectrum disorders and depression). An essential mechanism for modulating inhibitory synaptic transmission and plasticity is altering the postsynaptic abundance of GABA$_A$Rs[1]. Long-term potentiation and inhibition of GABAergic

transmission have been associated with increased or decreased GABA$_A$R synaptic abundance, respectively[2,3].

Like other postsynaptic receptors, GABA$_A$Rs at the cell surface undergo lateral diffusion and require postsynaptic scaffolds to stabilize their synaptic enrichment[2-5]. In mammals, GABA$_A$R clustering and anchoring are mediated by complex inhibitory postsynaptic scaffolds major consist of gephrin, collybistin, Neuroligin-2, and LHFPL4/GARLH4[5-14]. Previous research has shown that the GABA$_A$Rs are

[1]School of Life Science and Technology, ShanghaiTech University, Shanghai 201210, China. [2]Institute of Neuroscience, CAS Center for Excellence in Brain Science and Intelligence Technology, Chinese Academy of Sciences, Shanghai 200031, China. [3]University of Chinese Academy of Sciences, Beijing 100190, China. [4]Queensland Brain Institute, Clem Jones Centre for Ageing Dementia Research (CJCADR), The University of Queensland, Brisbane, QLD 4072, Australia. [5]College of Life Science and Technology, Huazhong University of Science and Technology, Wuhan 430074, China. [6]These authors contributed equally: Yue Hao, Haowen Liu. ✉e-mail: tongxj@shanghaitech.edu.cn

stabilized by distinct scaffolds at *C. elegans* neuromuscular junctions (NMJs), including the synaptic adhesion molecule NLG-1/Neuroligin, and the FERM domain-containing protein FRM-3[15–18]. The postsynaptic GABA$_A$Rs and inhibitory postsynaptic currents are eliminated in double mutants lacking both NLG-1 and FRM-3[17].

Besides the postsynaptic scaffolds, the presynaptic neurons also play essential roles in positioning and clustering postsynaptic receptors. During synaptogenesis, the innervation by presynaptic neurons releases anterograde signals to induce receptor aggregation and clustering in the post-junctional membranes. For example, at mammalian cholinergic synapses, cholinergic motor neurons release extracellular proteoglycan agrin to recruit acetylcholine receptors to postsynaptic membranes[19–21]. Other reports have shown that GABAergic motor neurons secrete the ADAMTS-like extracellular protein MADD-4B/Punctin, which promotes the localization of GABA$_A$Rs at inhibitory synapses at NMJs in *C. elegans*[15,16]. In addition, during the induction of activity-dependent synaptic plasticity, the excitation of the presynaptic neurons could trigger the recruitment of postsynaptic receptors and require anterograde signals released by presynaptic neurons. The transsynaptic recruitment of receptors has been extensively studied at excitatory synapses[22–27]. However, how the GABAergic neurons trigger anterograde signals to recruit GABA$_A$Rs remains unclear.

Calcium/calmodulin-dependent protein kinase II (CaMKII) is a serine/threonine-specific protein kinase activated by the Ca$^{2+}$/Calmodulin and functions as a ubiquitous mediator of cellular Ca$^{2+}$ signals[28]. CaMKII is well known as an essential component of postsynaptic density (PSD) proteins at excitatory synapses[29], and necessary for NMDA receptor-dependent long-term potentiation (LTP)[30–36]. At GABAergic synapses, postsynaptic CaMKII has been reported to phosphorylate GABA$_A$Rs within the TM3-4 domain to regulate receptor insertion at the cell surface[37–41]. CaMKII is required for the induction of rebound potentiation in Purkinje neurons[42–44] and is also required for the moderate N-methyl-D-aspartate receptor (NMDAR)-activating stimuli-induced long-term potentiation of inhibition (iLTP) in hippocampal neurons[2,45]. Besides that, it has been known for a long time that CaMKII is expressed and functions on the presynaptic side[35,46–51]. However, whether the presynaptic CaMKII could transsynaptically recruit postsynaptic receptors and be involved in inhibitory synaptic transmission and plasticity remain elusive.

Here, we utilize the *C. elegans* NMJs as a model to study how presynaptic neurons trigger anterograde signals to recruit GABA$_A$Rs at inhibitory synapses. We found that UNC-43, the *C. elegans* ortholog of CaMKII, is required for GABA$_A$Rs recruitment and modulates inhibitory synaptic transmission. Experiments using multiple reporter fusion constructs showed that UNC-43 functions at GABAergic motor neurons to recruit GABA$_A$Rs in the same pathway with NLG-1, but not FRM-3. Next, we demonstrated that UNC-43 promotes presynaptic MADD-4B secretion and cell adhesion molecule NRX-1α/Neurexin GABAergic motor neuron surface delivery. Together, MADD-4B and NRX-1α recruit postsynaptic NLG-1 and stabilize GABA$_A$Rs. Further, we confirmed that the activity-dependent plasticity in the inhibitory synapses requires the presynaptic UNC-43, MADD-4B, and NRX-1α, and is mediated by the NLG-1-stabilized GABA$_A$Rs. Collectively, our work elucidates how presynaptic neurons transsynaptically recruit postsynaptic receptors during synaptogenesis and activity-dependent plasticity.

## Results

### Presynaptic UNC-43/CaMKII regulates postsynaptic GABA$_A$Rs abundance at GABAergic synapses

To study the function of CaMKII on inhibitory synaptic transmission, we took advantage of the neuromuscular junction (NMJ) as a synaptic model. In *C. elegans*, the majority of GABAergic neurons in the nervous system are inhibitory motor neurons that innervate body-wall

muscles[52–55]. At the nerve cord, muscle arms elongate and form synapses with GABAergic axon terminals. As a result, the *C. elegans* body-wall muscles receive direct synaptic inputs from both cholinergic and GABAergic motor neurons[56]. To visualize endogenous GABA$_A$Rs, we used a similar strategy as the previous report by inserting a tagRFP coding sequence after the signal peptide sequence of the *unc-49* gene that encodes GABA$_A$R in *C. elegans* using the CRISPR-cas9 genome editing system[18]. The tagRFP can label all three GABA$_A$R subunits: UNC-49A, UNC-49B, and UNC-49C (Fig. 1a). To test whether the tagRFP fused GABA$_A$Rs fold and function well, we performed aldicarb assays to study whether the inhibitory synaptic transmission at NMJs is altered in this tagRFP-tagged GABA$_A$Rs imaging strain; very briefly, the cholinesterase inhibitor aldicarb prevents acetylcholine breakdown, and the attendant acetylcholine accumulation at synapses leads to muscle over-excitation and worm paralysis. Fundamentally, this assay can monitor the alteration of excitatory and inhibitory synaptic transmission[57]. During the 120 minutes of exposure to aldicarb, we found that the paralysis rate of tagRFP knock-in worms did not differ from the wild type (Supplementary Fig. 1). In contrast, the *unc-49* null mutants showed accelerated paralysis compared to the wild-type worms, thus implying impairment of inhibitory synaptic transmission at NMJs (Supplementary Fig. 1) and supporting that our tagRFP-tagged GABA$_A$Rs are functional.

To determine whether UNC-43/CaMKII participates in regulating GABA$_A$R abundance, we used the *unc-43(js125)* null allele[50,58], which deletes 17 kb of sequence positioned downstream of the promoter and the 10$^{th}$ exon and lacks most of the sequence encoding UNC-43 (Fig. 1b). Note that in *C. elegans*, UNC-43 encodes the sole CaMKII. Quantitative imaging analysis indicated that the GABA$_A$R puncta fluorescence intensity (puncta peak fluorescence to cord fluorescence ratio, peak-to-cord) in the *unc-43* mutants was decreased by around 35% compared with wild type (Fig. 1c), suggesting that UNC-43 does affect GABA$_A$R synaptic abundance. We also examined whether UNC-43 exerts a similar function for cholinergic receptors by measuring the synaptic abundance of two acetylcholine receptors L-AchR (UNC-29) and N-AchR (ACR-16) in the *unc-43* mutant animals. However, neither of them presented decreased fluorescent intensities compared to the wild type (Supplementary Fig. 2). Together, these findings support that UNC-43 is specifically involved in maintaining GABA$_A$R abundance at inhibitory synapses.

UNC-43 is expressed in both pre- and postsynapses at excitatory and inhibitory synapses[35,46–51]. At *C. elegans* NMJs, the excitatory motor neurons innervate and contract muscles, and also synapses onto the inhibitory motor neurons to relax the contralateral muscles to generate the sinusoidal movement[53]. To examine where UNC-43 functions to stabilize the postsynaptic GABA$_A$Rs, we restored UNC-43 expression in GABAergic motor neurons (under the *unc-25* promoter), in cholinergic motor neurons (under the *unc-17* promoter), or in body-wall muscles (under the *myo-3* promoter) (all in the *unc-43* mutant background). We found that the decreased synaptic abundance of GABA$_A$Rs in the *unc-43* mutant was partially rescued by either UNC-43D or UNC-43G isoform[59–61] in GABAergic motor neurons; no rescue was observed with the cholinergic neuron or body-wall muscle groups (Fig. 1c and Supplementary Fig. 3). All of the data supports that UNC-43 functions in the GABAergic motor neurons to stabilize postsynaptic GABA$_A$Rs.

Since we observed a partial rescue of GABA$_A$Rs recruitment defects in *unc-43* mutants by expressing UNC-43 in the GABAergic motor neurons, therefore, we restored UNC-43 (UNC-43D isoform) expression in pan-neurons (under *rab-3* or *unc-43* promoter), in both GABAergic and cholinergic motor neurons (under *unc-25* promoter +*unc-17* promoter), and in both GABAergic motor neurons and muscle cells (under *unc-25* promoter+*myo-3* promoter) in the *unc-43* mutants, and we found that the GABA$_A$Rs puncta fluorescence defects in the *unc-43* mutants can be fully rescued by expressing UNC-43 in

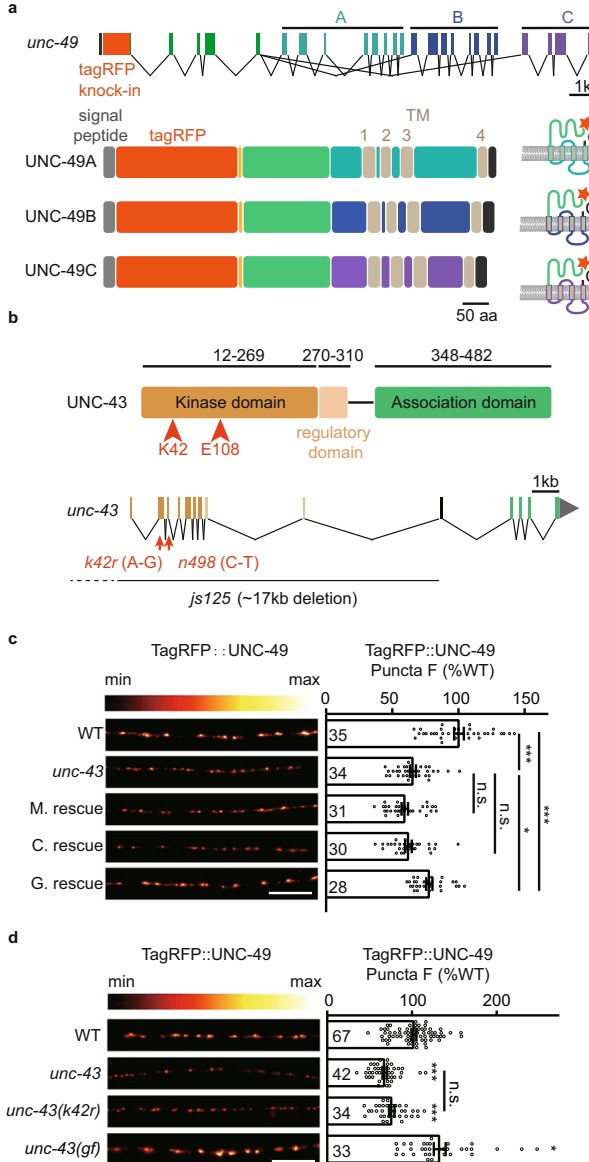

pan-neurons or in both GABAergic and cholinergic motor neurons, but not with both GABAergic motor neurons and muscle cells groups (Supplementary Fig. 3). These data suggest that the cholinergic UNC-43 is also involved in GABA$_A$Rs recruitment with unknown mechanisms, and it depends on the GABAergic UNC-43.

As a calcium/calmodulin-dependent protein kinase, we next studied whether its kinase activity is required for UNC-43-mediated recruitment of GABA$_A$Rs. We generated a kinase-dead mutant *unc-43(k42r)* using CRISPR-cas9 and analyzed GABA$_A$R abundance. The GABA$_A$R fluorescence intensity was reduced by about 25% in *unc-43(k42r)* (Fig. 1d), indicating that UNC-43's kinase activity is involved in the recruitment of GABA$_A$Rs. Further, we also analyzed the GABA$_A$R synaptic abundance in the *unc-43(n498)* gain-of-function allele that bears a constitutively activating mutation E108K in the active site core[62]. The GABA$_A$R fluorescence intensity was significantly increased in the *unc-43* mutant (Fig. 1d), supporting the conclusion that activation of UNC-43 promotes GABA$_A$Rs recruitment.

## UNC-43/CaMKII is required for the GABAergic synaptic localization of GABA$_A$Rs

To further investigate whether the remaining GABA$_A$Rs in the *unc-43* mutant are correctly localized at GABAergic synapses, we labeled the GABAergic synapses by fusing the endogenous UNC-57/Endophilin in GABAergic motor neurons with GFP in the tagRFP-tagged GABA$_A$Rs strain using a split GFP complementation system[63,64] (Fig. 2a), and calculated the colocalization coefficient between UNC-57 and GABA$_A$Rs (Fig. 2b). We found that the colocalization coefficient in the *unc-43* mutant was significantly decreased compared with the wild type ($0.7770 \pm 0.0143$ for wild type vs. $0.4848 \pm 0.0316$ for *unc-43*) (Fig. 2c, d). Further, GABA$_A$R mislocalization could be rescued by restoring UNC-43 expression in the GABAergic motor neurons (Fig. 2c, d), indicating that presynaptic UNC-43 is required for the GABAergic synaptic localization of GABA$_A$Rs.

## Presynaptic UNC-43 modulates inhibitory synaptic transmission at NMJs

To determine whether the observed GABA$_A$Rs synaptic recruitment defect in the *unc-43* mutants caused impairment of GABAergic synaptic transmission, we patch-clamped body-wall muscles and recorded spontaneous miniature inhibitory postsynaptic currents (mIPSCs). Consistent with the previous report[50], we observed a dramatic decrease in mIPSC frequency in the *unc-43* mutants (Fig. 3a, b). Besides the mIPSC frequency, the mISPC amplitude was also significantly decreased in the *unc-43* mutants (Fig. 3a, c), implying a defect in postsynaptic GABA$_A$Rs. Further, the decreased mIPSC frequency and amplitude could also be partially recovered by expressing UNC-43 in GABAergic neurons but not by expressing UNC-43 in muscles or cholinergic neurons (Fig. 3a–c).

To further confirm UNC-43 functions at GABAergic motor neurons to modulate inhibitory synaptic transmission, we generated worms with a conditional *unc-43* deletion in GABAergic motor neurons, or in muscle cells. Briefly, we inserted two LoxP sites at the *unc-43* promoter and the first intron, respectively, and drove CRE recombinase expression in the GABAergic motor neurons (under *unc-25* promoter) and body-wall muscles (under *myo-3* promoter) (Fig. 3d). We found that *unc-43* deletion at the GABAergic motor neurons mimics the phenotype of the *unc-43* knockout mutants, showing a significant decrease in mIPSC frequency and amplitude (Fig. 3e–g). Note that there were no obvious mIPSC frequency or amplitude defects in the strains with *unc-43* deletion in the body-wall muscles (Fig. 3e–g). These results confirm that UNC-43 functions at GABAergic motor neurons to support inhibitory synaptic transmission.

To investigate whether the expression and/or surface delivery of GABA$_A$Rs are compromised in the *unc-43* mutants, we recorded the 0.2 mM and 0.5 mM GABA-evoked currents, which were unaltered in

**Fig. 1 | Presynaptic UNC-43/CaMKII regulates postsynaptic GABA$_A$R abundance at GABAergic synapses. a** Schematic representation of the *xj1024* locus. A tagRFP coding sequence was inserted before the first exon shared by UNC-49A, UNC-49B, and UNC-49C. TagRFP consequently labels the UNC-49A, UNC-49B, and UNC-49C proteins. **b** The protein structure of UNC-43 is shown on top. The *js125* mutation has a 17 kb deletion (upstream of the promoter to the 10th exon) that deletes most of the coding sequences. The kinase-dead mutant *unc-43 (k42r)* has a point mutation and causes the K42R coding variant. The gain-of-function *unc-43 (gf) (n498*, gain of function) mutant has a point mutation that causes single amino acid substitution E108K. **c** The puncta fluorescence intensity marked by TagRFP-UNC-49 (under *unc-49* own promoter) in dorsal nerve cord axons was decreased in the *unc-43* mutants. This defect was rescued by transgenic expression of UNC-43 in GABAergic motor neurons (G. rescue), but not by UNC-43 expression in cholinergic neurons (C. rescue) or in body-wall muscles (M. rescue). Representative images (left, scale bar 10 μm, Pseudo-color: Red Hot) and mean puncta intensities +/− SEM (right, Ns represent the number of animals tested) are shown. One-way ANOVA with post-hoc Bonferroni's multiple comparison test. *$p < 0.05$, ***$p < 0.001$, n.s. not significant. **d** GABA$_A$R fluorescence intensity is decreased in the *unc-43 (k42r)* kinase-dead mutants, and increased in the *unc-43(gf) (n498*, gain of function) mutants. Representative images (left, scale bar 10 μm, Pseudo-color: Red Hot) and mean puncta intensities +/- SEM (right, Ns represent the number of animals tested) are shown. Kruskal–Wallis test with post-hoc Dunn's test. *$p < 0.05$, ***$p < 0.001$, n.s. not significant. For **c**, **d**, source data are provided as a Source Data file.

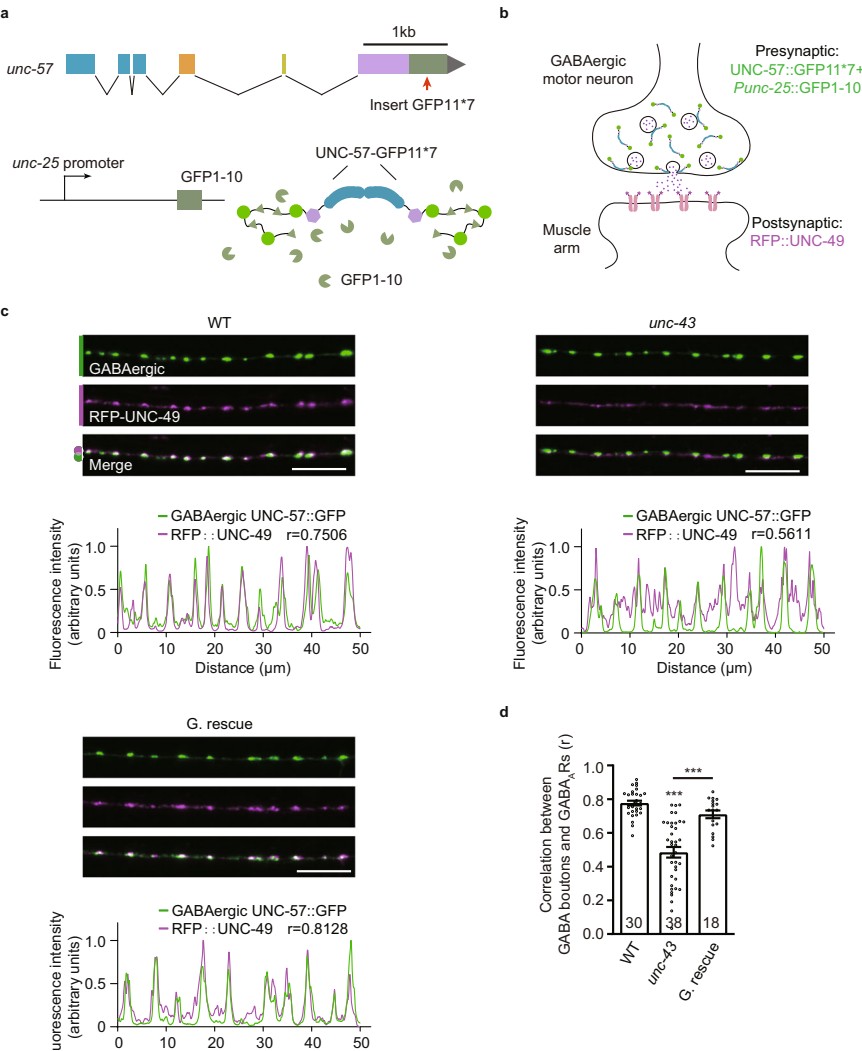

**Fig. 2 | Presynaptic UNC-43/CaMKII is required for the GABAergic synaptic localization of GABA_ARs. a, b** Schematic illustration of labeling UNC-57 at the GABAergic motor neurons by split GFP complementary system. Seven copies of the split GFP11 were inserted into the C-terminal of *unc-57* genomic loci by CRISPR-Cas9 system. The split GFP1-10 was expressed in GABAergic motor neurons by *unc-25* promoters. **c, d** The colocalization coefficient between GABA_ARs and GABAergic synaptic boutons was decreased in the *unc-43* mutants; this was rescued by transgenic expression of UNC-43 in GABAergic motor neurons (G.

rescue). Pearson's correlation coefficients between the intensities of GABAergic bouton marker UNC-57 (green, labeled by UNC-57-split GFP under *unc-25* promoter) and postsynaptic GABA_ARs (magenta) were used to assess the localization of GABA_ARs at inhibitory synapses. Representative images (**c**-top, scale bar, 10 μm), corresponding line scan curve (**c**-bottom), and mean Pearson's correlation coefficients +/− SEM (**d**, Ns represent the number of animals tested) are shown. Kruskal−Wallis test with post-hoc Dunn's test. ***$p < 0.001$. Source data are provided as a Source Data file.

*unc-43* mutants (Fig. 3h–i and Supplementary Fig. 4). Thus, the mIPSC amplitude defect in the *unc-43* mutants is unlikely to be caused by decreased bulk expression and surface delivery of GABA_ARs. A previous study reported a minor decrease in response to pressure-applied GABA (0.2 mM) in the *unc-43* mutant[50]. This discrepancy likely arises from the different protocols of recording. These results verified that UNC-43 is required for both GABA_AR synaptic recruitment and inhibitory synaptic transmission.

To rule out the possibility that the decreased postsynaptic GABA_ARs in the *unc-43* mutant is the secondary result of synaptic structural defects, we measured the puncta fluorescence and intensity of endogenous UNC-57/Endophilin in GABAergic motor neurons in the *unc-43* mutant. We observed no significant decrease in puncta fluorescence intensities and densities between wild-type and *unc-43* mutants (Supplementary Fig. 5a–c). Further, we also labeled the endogenous UNC-2/CaV2 in the GABAergic motor neurons with GFP by split GFP complementation system[65]. Both the

fluorescence intensity and density of UNC-2-GFP fusion protein were not altered in the *unc-43* mutants compared to the wild type (Supplementary Fig. 5d–f). These results suggested that the GABAergic synapse structure was unaltered.

We also ruled out the possibility that the decreased postsynaptic GABA_ARs in the *unc-43* mutants result from diminished synaptic GABA transmission or neuropeptide release, as both a previous report and our data have demonstrated that the synaptic abundance of GABA_AR was not decreased in the *unc-25* mutants (which lack GABA biogenesis), *snb-1* mutants, *unc-13* mutants (which lack synaptic vesicle release), *unc-31* mutants (which lack dense-core vesicle release), and *egl-3* (which lack neuropeptide maturation) (Supplementary Fig. 6a–c)[66]. Further, deletion of *unc-43* causes a reduction of GABA_AR fluorescence in both *snb-1* and *unc-25* mutants (Supplementary Fig. 6a–b), indicating UNC-43 recruits GABA_ARs independent of GABAergic neurotransmission. The increase of GABA_ARs puncta fluorescence in the *unc-25* mutants may be caused by synaptic

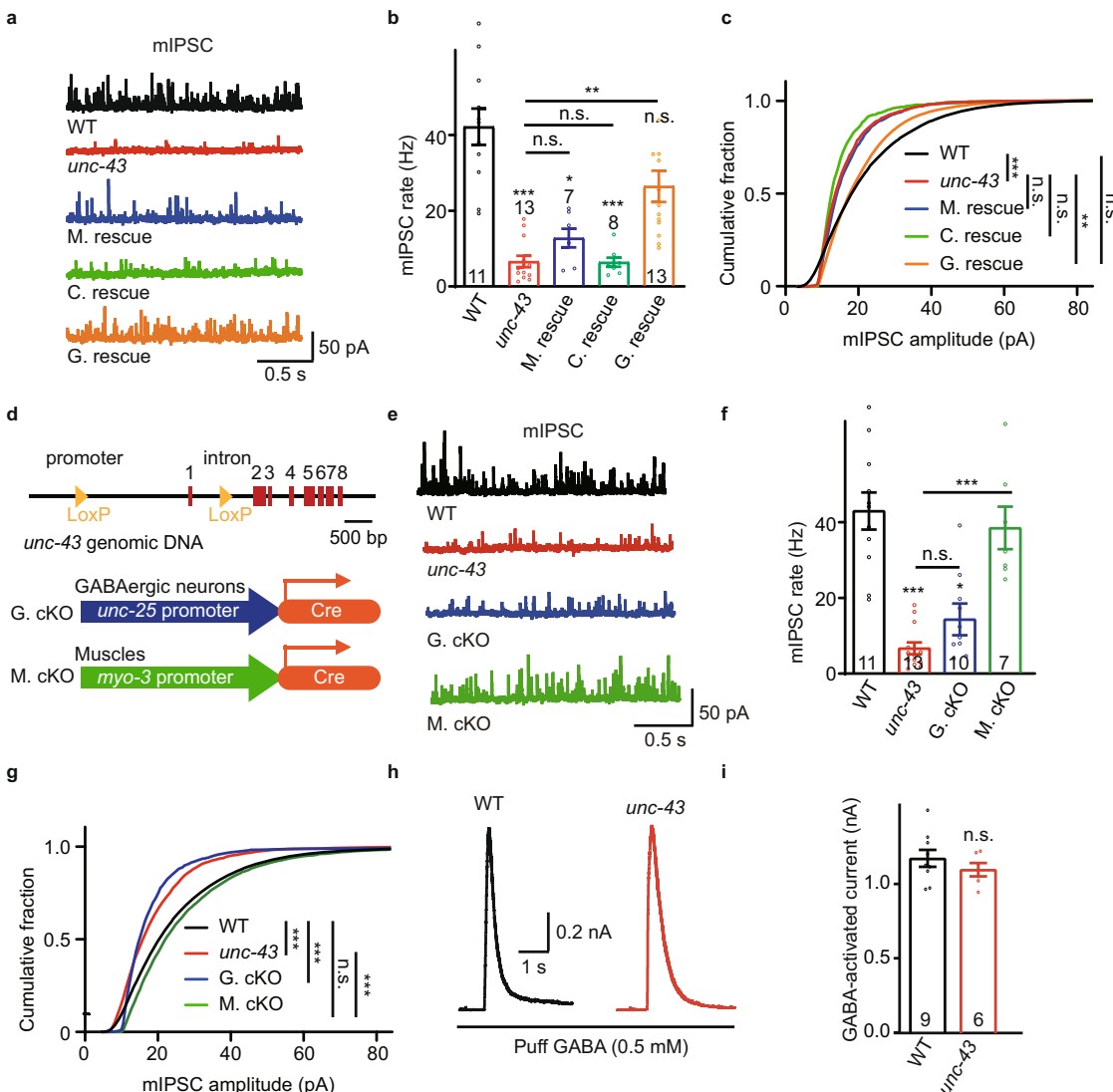

**Fig. 3 | Presynaptic UNC-43/CaMKII modulates inhibitory synaptic transmission. a–c** Endogenous inhibitory synaptic transmission was assessed by recording mIPSCs from body-wall muscles. The defects of mIPSC in the *unc-43* mutants were rescued by transgenic expression of UNC-43 in GABAergic motor neurons (G. rescue), but not by UNC-43 expression in cholinergic neurons (C. rescue) or in body-wall muscles (M. rescue). Representative mIPSC traces (**a**), mIPSC rates (**b**, Data are presented as mean values +/− SEM, Ns represent the number of animals tested.), and the cumulative fraction of mIPSC amplitude (**c**) are shown. Kruskal–Wallis test with post-hoc Dunn's multiple comparisons. *$p < 0.05$, **$p < 0.01$, ***$p < 0.001$, n.s. not significant. **d** Schematic for construction of the *unc-43* conditional deletion lines. Two LoxP sites were inserted into the genomic loci (at the promoter and the first intron of UNC-43 K11E8.1a to K11E8.1l) by CRISPR/Cas9, the minigene encoding Cre recombinase under *unc-25* and *myo-3* promoters were

used to knock out *unc-43* gene in GABAergic motor neurons (G.cKO) and muscle cells (M.cKO), respectively. **e–g** Endogenous inhibitory synaptic transmission was assessed by recording mIPSCs from body-wall muscles of *unc-43* conditional knockout animals. Representative mIPSC traces (**e**), mIPSC rates (**f**, Data are presented as mean values +/− SEM, Ns represent the number of animals tested.), and the cumulative fraction of mIPSC amplitude (**g**) are shown. Kruskal–Wallis test with post-hoc Dunn's multiple comparisons. *$p < 0.05$, ***$p < 0.001$, n.s. not significant. **h, i** The GABA-evoked currents in the *unc-43* mutants were comparable to that in wild-type animals (WT). Representative responses (**h**) and current amplitude (**i**, Data are presented as mean values +/− SEM, Ns represent the number of animals tested.) are shown. Two-tailed and unpaired Student's *t*-test. n.s. not significant. For **b, c, f, g, i**, source data are provided as a Source Data file.

homeostasis. Further, GABA$_A$Rs recruitment is not modulated by the BK channel SLO-1[50] (Supplementary Fig. 6b).

## UNC-43 recruits postsynaptic NLG-1 to stabilize GABA$_A$Rs

Our previous work reported that the postsynaptic GABA$_A$Rs are recruited by two distinct scaffolds FRM-3 and NLG-1/neuroligin[15–18]. To investigate whether UNC-43 functions in a single pathway with FRM-3 or with NLG-1 to stabilize GABA$_A$Rs, we generated *unc-43; frm-3* and *unc-43; nlg-1* double mutants and then analyzed GABA$_A$R fluorescence intensity at synapses. We observed a further reduction of GABA$_A$R puncta fluorescence intensity in the *unc-43; frm-3* double mutants compared to the *frm-3* and *unc-43* single mutants (Fig. 4a). However,

there was no difference in GABA$_A$R fluorescence intensity between the *unc-43; nlg-1* double mutants and the *nlg-1* single mutant worms (Fig. 4a). Besides, Super-resolution microscopy studies by Sora mode showed that the GABA$_A$Rs clustering was significantly decreased in both the *unc-43* mutants and the *nlg-1* mutants, but not in the *frm-3* mutants (Supplementary Fig. 7). Further, the increase of GABA$_A$R fluorescence intensity by *unc-43(n498)* gain-of-function mutation was eliminated in the *nlg-1* mutant, but not in the *frm-3* mutant (Fig. 4b), lending more support to the conclusion that UNC-43 functions in a single pathway with NLG-1 to recruit GABA$_A$Rs.

Next, we tested whether UNC-43 may somehow regulate the postsynaptic localization of NLG-1. We expressed the C-terminal

GFP-tagged NLG-1 (under the *myo-3* promoter) in the muscle cells of wild-type and *unc-43* mutant worms. The GABA$_A$Rs recruitment defects of *nlg-1* mutants were rescued by the transgene expression of the C-terminal GFP-tagged NLG-1 in the muscle cells (Supplementary Fig. 8). We detected a dramatic decrease in signal intensity for the NLG-1 puncta in the *unc-43* mutants (Fig. 4d). Further, we found that this NLG-1 synaptic abundance defect was rescued by restoring UNC-43 expression in GABAergic motor neurons (Fig. 4d), indicating pre-synaptic UNC-43 is required for NLG-1 postsynaptic abundance. It is worth noting that the postsynaptic localization of FRM-3 was unaltered in the *unc-43* mutants (Fig. 4c, Supplementary Fig. 8). These results suggested the possibility that UNC-43/CaMKII stabilizes synaptic GABA$_A$Rs by recruiting postsynaptic NLG-1.

### UNC-43 recruits postsynaptic NLG-1 and stabilizes GABA$_A$Rs requiring both NRX-1α and MADD-4B

Previous studies have shown that the ADAMTS-like secretion protein MADD-4B/Punctin can be secreted by GABAergic motor neurons, and showed that MADD-4B/Punctin acts partially redundantly with pre-synaptic NRX-1α/Neurexin to recruit postsynaptic NLG-1 at GABAergic synapses[15,16]. Consistent with previous reports, we found that *madd-4b* and *nrx-1* mutations cause an additive defect of NLG-1 synaptic locali-zation (Fig. 5a); specifically, we detected a further decrease in NLG-1 puncta fluorescence intensity in the *madd-4b; nrx-1* double mutants compared to either of the single mutants (Fig. 5a). In this context, we test whether UNC-43 may function to recruit NLG-1 in a single pathway with MADD-4B and/or NRX-1α. We analyzed the NLG-1 GABAergic synaptic abundance in the *madd-4b; nrx-1; unc-43* triple mutants: the *unc-43* deletion did not cause any further reduction in NLG-1 puncta fluorescence intensity as compared to the *madd-4b; nrx-1* double mutants (Fig. 5a). Together, these results support that a single genetic pathway that includes UNC-43, MADD-4B, and NRX-1α is responsible for the recruitment of postsynaptic NLG-1.

We then investigated whether UNC-43's recruitment of GABA$_A$Rs requires MADD-4B and NRX-1α. Consistent with previous reports[15], we found that MADD-4B and NRX-1α function redundantly to recruit GABA$_A$Rs: there was a further decrease in the fluorescence intensity of GABA$_A$Rs fusion protein puncta in the *madd-4b; nrx-1* double mutants as compared to both the *madd-4b* and *nrx-1* single mutants (Fig. 5b). Here we observed a less severe defect of GABA$_A$Rs recruitment in the *madd-4b;* and *nrx-1* mutants than in previous reports[15,16], it may be caused by different strategies to label GABA$_A$Rs. Note that the deletion of *unc-43* further decreases the GABA$_A$R puncta fluorescence intensity in the *madd-4b* and *nrx-1* single mutants (Fig. 5b), but not in the *madd-4b; nrx-1* double mutants (Fig. 5b). Further, *unc-43(n498)* gain-of-function mutation is not able to increase the GABA$_A$R puncta fluores-cence intensity in the *madd-4b; nrx-1* double mutants (Fig. 5c), indi-cating that UNC-43 recruits postsynaptic GABA$_A$Rs requiring both MADD-4B and NRX-1α.

### UNC-43 promotes MADD-4B/Punctin secretion and NRX-1α/Neurexin GABAergic motor neuron surface delivery

Since UNC-43 requires both MADD-4B and NRX-1α to recruit post-synaptic NLG-1, it is possible that UNC-43 regulates MADD-4 and NRX-1α localization. To study MADD-4B secretion, we expressed a C-terminal GFP-tagged MADD-4B fusion in GABAergic neurons (under the *unc-25* promoter), and the MADD-4B-GFP fusion protein is able to rescue the GABA$_A$Rs recruitment defects in the *madd-4b* mutants (Supplementary Fig. 8). A MADD-4B-GFP fluorescent signal was evident in both the nerve cords and coelomocytes. Coelomocytes are sca-venger cells that endocytose proteins secreted into the body cavity[67]. Accordingly, the GFP signal in the coelomocytes indicates the secreted MADD-4B, while the fluorescent signal from the dorsal cord indicates the MADD-4B that is retained at the motor neuron axon terminals.

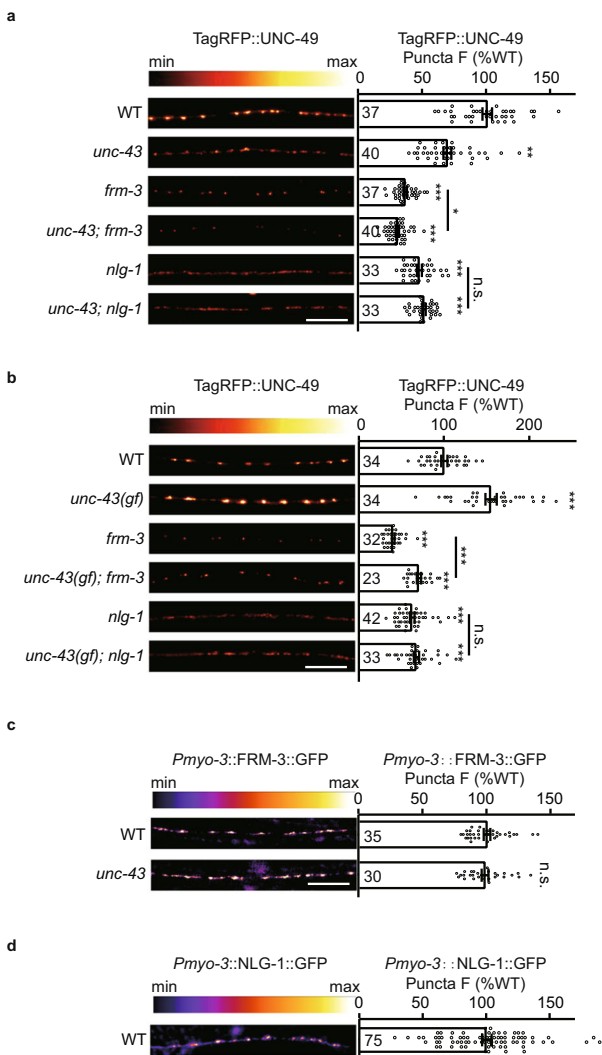

**Fig. 4 | UNC-43/CaMKII acts through a single pathway with NLG-1/neuroligin to recruit GABA$_A$Rs. a** The tagRFP-UNC-49 puncta fluorescence intensities in the *unc-43; nlg-1* double mutants were comparable to the *nlg-1* single mutants. Repre-sentative images (left, scale bar 10 μm, Pseudo-color: Red Hot) and mean puncta intensities +/− SEM (right, Ns represent the number of animals tested) are shown. Kruskal–Wallis test with two-stage linear step-up procedure of Benjamini, Krieger, and Yekutieli correction for multiple comparisons. *$p < 0.05$, **$p < 0.01$, ***$p < 0.001$, n.s. not significant. **b** The tagRFP-UNC-49 puncta fluorescence inten-sities in the *unc-43 (gf); nlg-1* double mutants were comparable to the *nlg-1* single mutants. Representative images (left, scale bar 10 μm, Pseudo-color: Red Hot) and mean puncta intensities +/− SEM (right, Ns represent the number of animals tested) are shown. One-way ANOVA with two-stage linear step-up procedure of Benjamini, Krieger, and Yekutieli correction for multiple comparisons. ***$p < 0.001$, n.s. not significant. **c** UNC-43 does not regulate postsynaptic FRM-3 localization. FRM-3-GFP fluorescence intensities in the body-wall muscles in wild-type and *unc-43* mutants were shown. Representative images (left, scale bar 10 μm, Pseudo-color: Fire) and mean puncta intensities +/− SEM (right, Ns represent the number of animals tested) are shown. Two-tailed and unpaired Student's *t*-test. n.s. not sig-nificant. **d** UNC-43 functions in the GABAergic neurons to stabilize postsynaptic NLG-1 localization. NLG-1-GFP fluorescence intensities in the body-wall muscles in wild-type and *unc-43* mutants were shown. Representative images (left, scale bar 10 μm, Pseudo-color: Fire) and mean puncta intensities +/− SEM (right, Ns repre-sent the number of animals tested) are shown. Kruskal–Wallis test with post-hoc Dunn's test. ***$p < 0.001$, n.s. not significant. For **a**–**d**, source data are provided as a Source Data file.

Compared to the wild type, there was a significant decrease of MADD-4B-GFP fluorescence intensity in coelomocytes of both the *unc-43* knockout and *unc-43(k42r)* kinase-dead mutants (Fig. 6a, b), and the MADD-4B signal in both *unc-43* mutants dorsal nerve cord was significantly increased (Fig. 6c). As a result, the decrease of MADD-4B-GFP fluorescence intensity in the coelomocytes and the increase of MADD-4B-GFP fluorescence intensity in the dorsal nerve cord as shown in the *unc-43* mutants indicate MADD-4B secretion by GABAergic motor neurons is partially blocked in the *unc-43* mutants, thus supporting that UNC-43 promotes MADD-4B secretion and it requires UNC-43's kinase activity.

Here we ruled out the possibility that the decreased MADD-4B-GFP fluorescence intensity in coelomocytes in *unc-43* mutants is caused by abnormal coelomocyte functions or general defect in pre-synaptic protein secretion, as the endocytosis of a constitutive GFP secreted from muscle cells or from the GABAergic motor neurons by coelomocytes was not affected by *unc-43* mutation (Supplementary Fig. 9).

To study whether UNC-43 regulates the surface delivery of NRX-1α in GABAergic motor neurons, we expressed a dual-tagged NRX-1α fusion protein by inserting pHluorin at the N-terminus and wrmScarlet at the C-terminus of NRX-1α under the *unc-25* promoter (Fig. 7a). pHluorin is a pH-sensitive version of GFP, and receptors with extracellular domain tagged pHluorin are fluorescent upon insertion in the plasma membrane to expose pHluorin in the extracellular environment (PH > 6). Acidification of the extracellular environment to pH 5.5 quenched all the fluorescence in both wild-type and *unc-43* mutants (Supplementary Fig. 10), suggesting that all of the fluorescent signals are from the pHluorin-NRX-1 α in the plasma membrane. We observed a dramatic decrease of the pHluorin fluorescent signal in both *unc-43* knockout and *unc-43 (k42r)* mutants (Fig. 7b, c, e), while the wrmScarlet fluorescence intensities were unaltered (Fig. 7b, d), indicating that the GABAergic motor neuron surface delivery of NRX-1α, but not its expression or axonal trafficking, was promoted by UNC-43.

## UNC-43/CaMKII-triggered anterograde signals are required for activity-dependent plasticity at GABAergic synapses

Previous research shows that UNC-43 is required for activity-dependent plasticity at PLM-AVA synapses by regulating synaptic AMPARs trafficking after presynaptic neuron excitation or silencing[30,33,68,69]. Thus, we tested whether UNC-43 is also required for activity-dependent plasticity at GABAergic synapses of NMJs. To study the activity-dependent plasticity of GABAergic synapses, we opto-genetically activated the GABAergic motor neurons by expressing a channelrhodopsin variant (ChIEF) in the GABAergic motor neurons (under *unc-25* promoter) and delivering pulsed blue-light excitation (Fig. 8a). After 30 min of excitation, we analyzed the GABA$_A$R synaptic abundance and found a significant increase of GABA$_A$R puncta fluorescence intensity compared with controls without all-trans retinal (ATR), and it lasted within 2 h (Fig. 8b, c). Besides, the colocalization coefficient between the presynaptic marker UNC-57 and GABA$_A$Rs was slightly but significantly increased (Fig. 8d, e). To study whether the increased GABA$_A$Rs recruitment upon GABAergic motor neuron excitation potentiates inhibitory synaptic transmission, we patch-clamped body-wall muscles and recorded mIPSCs. Both the mIPSCs frequency and amplitude were significantly increased in those animals with GABAergic motor neuron excited compared to controls without all-trans retinal (ATR) (Fig. 8f–h), supporting that the inhibitory synaptic transmission was also increased. This result indicates the existence of activity-dependent synaptic plasticity at GABAergic synapses.

Next, we studied whether GABAergic activity-dependent plasticity requires UNC-43. We detected no increase of GABA$_A$R puncta fluorescence at synapses afterward in the *unc-43* mutant animals, and restoring UNC-43 expression in GABAergic motor neurons rescued the activity-dependent increase of GABA$_A$R recruitment (Fig. 9a),

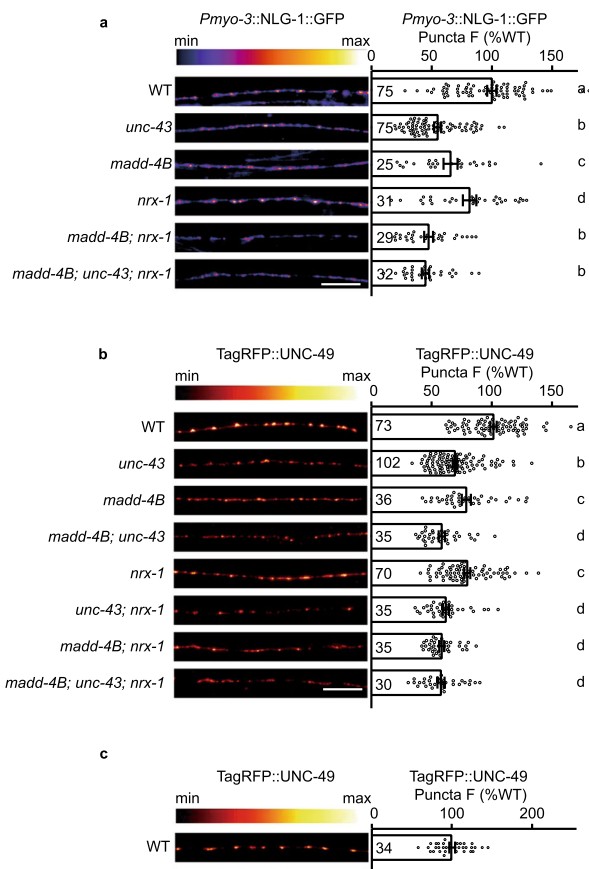

**Fig. 5 | UNC-43/CaMKII's recruitment of GABA$_A$Rs requires both MADD-4B and NRX-1α. a** UNC-43 functions in the same pathway with MADD-4B and NRX-1α to stabilize postsynaptic NLG-1 localization. NLG-1-GFP fluorescence intensities in the body-wall muscles in wild type and mutants were shown. Representative images (left, scale bar 10 μm, Pseudo-color: Fire) and mean puncta intensities +/− SEM (right, Ns represent the number of animals tested) are shown. The data for WT and *unc-43* are the same as in Fig. 4c. Kruskal–Wallis test with two-stage linear step-up procedure of Benjamini, Krieger, and Yekutieli correction for multiple comparisons. The scatter plotted data labeled with different letters are significantly different (*p* < 0.05). **b, c** Both MADD-4B and NRX-1α are required for UNC-43's recruitment of GABA$_A$Rs. TagRFP-UNC-49 fluorescence in dorsal nerve cords in wild type and mutants is shown. Representative images (left, scale bar 10 μm, Pseudo-color: Red Hot) and mean puncta intensities +/− SEM (right, Ns represent the number of animals tested) are shown. In **b**, Kruskal–Wallis test with two-stage linear step-up procedure of Benjamini, Krieger, and Yekutieli correction for multiple comparisons. The scatter plotted data labeled with different letters are significantly different (*p* < 0.05). In **c**, Kruskal–Wallis test with post-hoc Dunn's test. **p* < 0.01, ***p* < 0.001, n.s. not significant. For **a–c**, source data are provided as a Source Data file.

indicating that the presynaptic UNC-43 is required for the activity-dependent plasticity at GABAergic synapses. Besides, GABAergic activity-dependent plasticity was not eliminated in the *unc-43* gain-of-function mutants (Fig. 9a).

Further, we asked whether the presynaptic localization of UNC-43 is altered upon GABAergic motor neuron excitation. To visualize endogenous UNC-43 at GABAergic motor neurons, we utilized the split GFP complementary system[63,64]. We used CRISPR/Cas9 system to insert a sequence coding for seven GFP11 fragments at the N-terminus of UNC-43. In parallel, the GFP1-10 fragment was constitutively

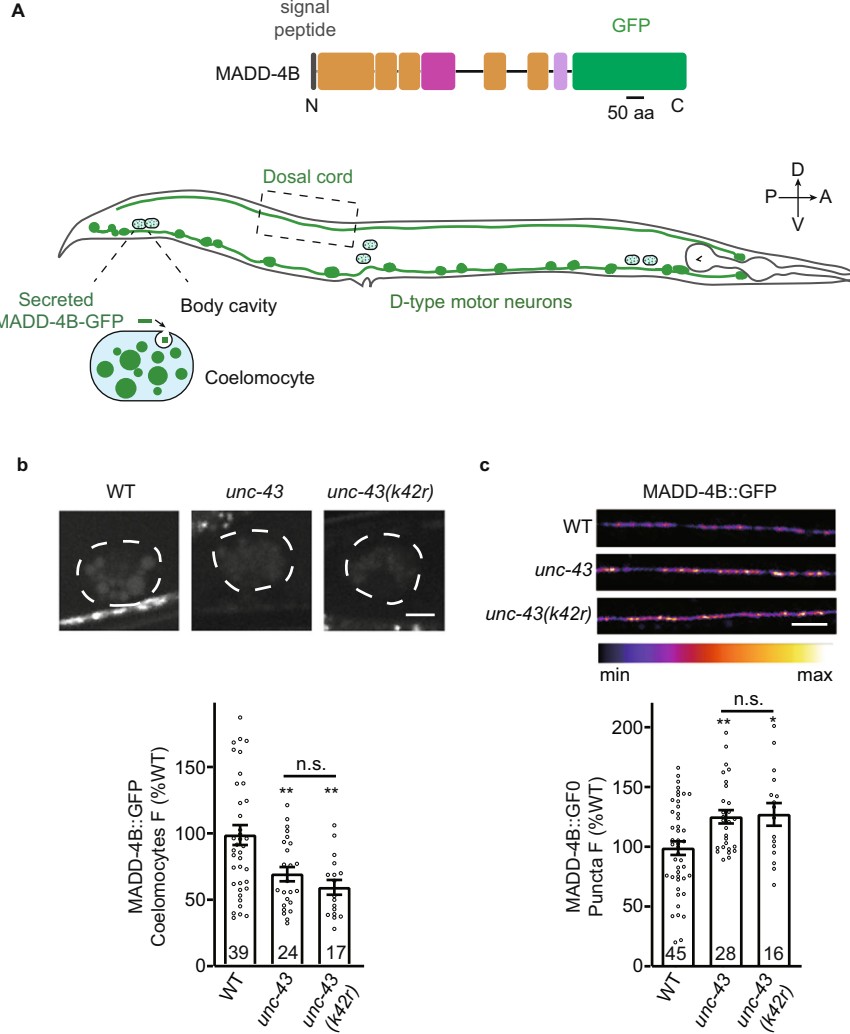

**Fig. 6 | UNC-43/CaMKII promotes MADD-4B/Punctin secretion. a** Schematic representation of C-terminal GFP-tagged MADD-4B fusion under *unc-25* promoter (top) and MADD-4B-GFP expressed in D-type motor neurons, meanwhile the secreted MADD-4B-GFP in body cavity was endocytosed by scavenger cell coelomocytes (bottom). **b** MADD-4B secretion by GABAergic motor neurons was decreased in the *unc-43* knockout and *unc-43 (k42r)* mutants. Secretion of MADD-4B was measured by analyzing GFP fluorescence intensities in the endolysosomal compartment coelomocytes. Representative images (top, scale bar 5 μm) and mean fluorescence intensities +/− SEM (bottom, Ns represent the number of animals tested) are shown. one-way ANOVA with post-hoc Bonferroni's multiple comparison test. **$p < 0.01$, n.s. not significant. **c** Puncta fluorescence of MADD-4B-GFP in the dorsal nerve cords was measured in the *unc-43* mutants. Representative images (top, scale bar 10 μm, Pseudo-color: Fire) and mean puncta intensities +/− SEM (bottom, Ns represent the number of animals tested) are shown. One-way ANOVA with post-hoc Bonferroni's multiple comparison test. *$p < 0.05$, **$p < 0.01$, n.s. not significant. For **b**, **c**, source data are provided as a Source Data file.

expressed in the GABAergic motor neurons under the control of the *unc-25* promoter (Fig. 9b) to label the endogenous localization of UNC-43 in GABAergic motor neurons. We observed a significant increase in the GFP fluorescent signal upon blue-light-stimulated GABAergic neuron excitation (Fig. 9c), supporting the conclusion that more UNC-43 are localized at the GABAergic motor neuron axon terminals upon GABAergic motor neuron stimulation.

Since UNC-43 is required for the increase of the synaptic abundance of GABA$_A$Rs after excitation of GABAergic motor neurons, linking with our finding that UNC-43 acts through a single pathway with NLG-1 to recruit GABA$_A$Rs, it is likely that the NLG-1-stabilized diffusing GABA$_A$Rs, rather than the FRM-3-stabilized immobilized GABA$_A$Rs, mediate the inhibitory activity-dependent plasticity. To test this hypothesis, we turned to the *nlg-1* and *frm-3* mutants. We observed an increase of GABA$_A$Rs recruitment after GABAergic motor neuron excitation in the *frm-3* mutants, but not in the *nlg-1* mutants (Fig. 9d), which supports our hypothesis that the diffusing GABA$_A$Rs stabilized by NLG-1 mediate activity-dependent plasticity at inhibitory synapses.

To study whether the UNC-43-participated activity-dependent plasticity requires MADD-4B and NRX-1α, we studied the activity-dependent plasticity in the *madd-4b; nrx-1* double mutants. Similar to in the *unc-43* mutants, the increase of GABA$_A$Rs postsynaptic abundance post GABAergic motor neuron excitation was blocked in the *madd-4b; nrx-1* double mutants (Fig. 9d). Further, the secretion of MADD-4B and the surface delivery of NRX-1α were both increased upon GABAergic neuron excitation (Supplementary Fig. 11), supporting the conclusion that MADD-4 and NRX-1α also participate in the activity-dependent plasticity at GABAergic synapses, and they are very likely to work in the single pathway with UNC-43.

## Discussion

In this study, we revealed how UNC-43/CaMKII participates in trans-synaptically recruiting GABA$_A$Rs at NMJs. We showed that the presynaptic UNC-43/CaMKII promotes MADD-4B secretion and delivery of the cell adhesion molecule NRX-1α to the surface of GABAergic motor neurons. MADD-4B and NRX-1α act as anterograde signals to recruit

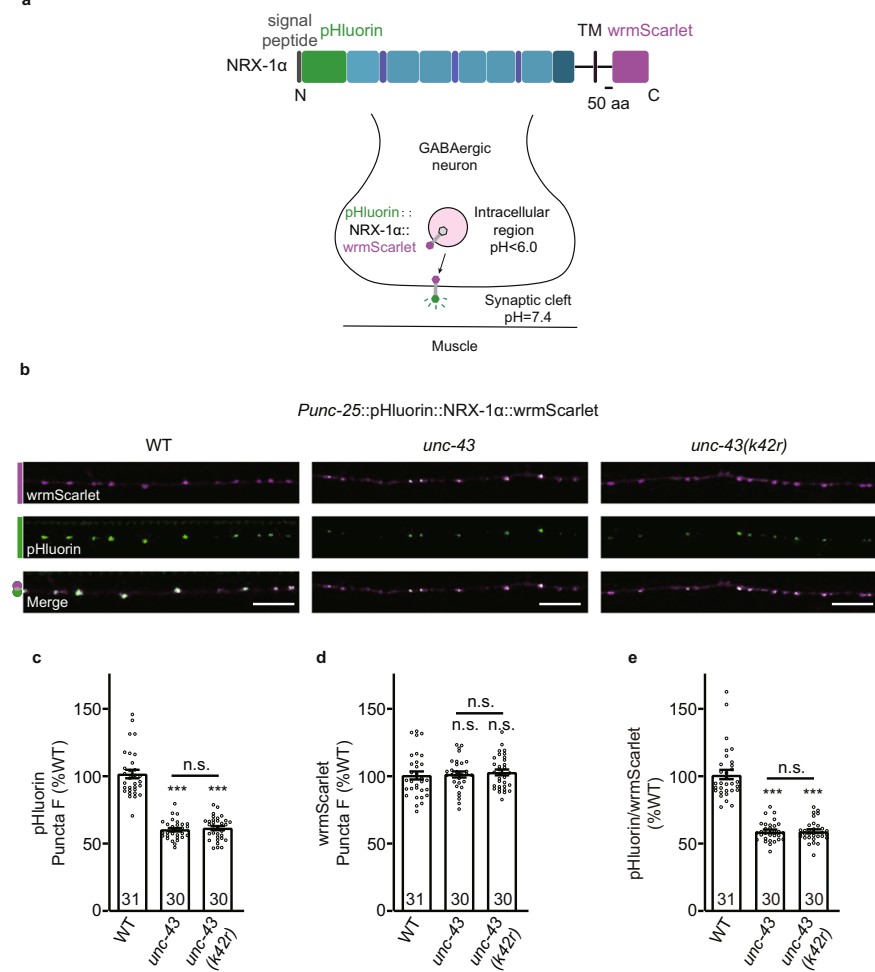

**Fig. 7 | UNC-43/CaMKII promotes NRX-1α/neurexin GABAergic motor neuron surface localization. a** Schematic representation of N-terminal pH-sensitive GFP pHluorin-tagged and C-terminal wrmScarlet-tagged NRX-1α fusion under *unc-25* promoter. pHluorin-NRX-1α are fluorescent upon insertion in the plasma membrane to expose pHluorin in the extracellular environment (pH>6). **b–e** The surface localization of NRX-1α is decreased in the *unc-43* mutants. pHluorin and wrmScarlet dual-labeled NRX-1α was expressed in GABAergic neurons. pHluorin-NRX-1α puncta (green) fluorescence in the dorsal nerve cords was decreased in the *unc-43*

mutants (**c**). NRX-1α-wrmScarlet puncta (magenta) fluorescence in the dorsal nerve cords was unaltered in the *unc-43* mutants (**d**). The averaged fluorescence intensity of pHluorin normalized to wrmScarlet fluorescence intensity in the *unc-43* mutants (**e**). Representative images (top, scale bar 10 μm) and mean puncta intensities +/− SEM (bottom, Ns represent the number of animals tested) are shown. One-way ANOVA with post-hoc Bonferroni's multiple comparison test for **c, d**, Kruskal–Wallis test with post-hoc Dunn's test for **e**. ***$p < 0.001$, n.s. not significant. For **c–e**, source data are provided as a Source Data file.

## The presynaptic function of UNC-43/CaMKII

postsynaptic NLG-1 and stabilize the GABA$_A$Rs at inhibitory synapses. We further demonstrated that the NLG-1-stabilized GABA$_A$Rs, but not the FRM-3-stabilized GABA$_A$Rs, mediate activity-dependent plasticity at GABAergic synapses, and experimentally confirmed that this mediation requires UNC-43, MADD-4B, and NRX-1α. These mechanistic insights about how presynaptic neurons transsynaptically recruit postsynaptic GABA$_A$Rs deepen our understanding about pre- and postsynaptic communications and inhibitory synaptic plasticity.

## The presynaptic function of UNC-43/CaMKII

Most previous studies of CaMKII have focused on its facilitation of AMPAR recruitment to enhance synaptic strength at postsynapses[30,33]. At inhibitory synapses, CaMKII is reported to phosphorylate GABA$_A$Rs and regulates their surface delivery[37–41]. In hippocampal neurons, the moderate N-methyl-D-aspartate receptor (NMDAR)-activating stimuli cause CaMKII selectively translocating to inhibitory synapses to phosphorylate GABA$_A$R β3S383 and recruit scaffold protein gephyrin, which promotes GABA$_A$Rs accumulation and immobilization at synapses[2,45,70]. Besides, an acute increase in neuronal activity of cultured hippocampal neurons also promotes the phosphorylation of

β3S383 by CaMKII[39]. Further, in cerebellar somatodendritic basket cell synapses, CaMKII is required for the rebound potentiation (RP) by phosphorylation of GABA$_A$Rs receptor[44].

However, many studies have indicated that CaMKII has functional impacts at presynapses[35,46–51]. CaMKII is enriched at presynaptic sites, and accounts for ~2% of the total synaptic vesicle protein amount[71]. Presynaptic injection of the membrane-impermeable CaMKII inhibitor peptide 281-309 was shown to block excitatory synaptic plasticity in cultured hippocampal neurons[47] and presynaptic CaMKII was implicated in synaptogenesis and synaptic transmission at NMJs of both *Drosophila* and *C. elegans*[46,50]. In the present study, we identified a transsynaptic GABA$_A$R recruitment function of presynaptic CaMKII, thus demonstrating CaMKII's involvement in inhibitory synaptic transmission. We also show that this transsynaptic recruitment activity is supported by MADD-4B secretion and NRX-1α surface delivery and propose the hypothesis that UNC-43-triggered anterograde signals are required for the activity-dependent plasticity at inhibitory synapses. Thus, the function of CaMKII in recruiting postsynaptic GABA$_A$Rs is conserved in species, and both presynaptic and postsynaptic CaMKII can recruit postsynaptic GABA$_A$Rs.

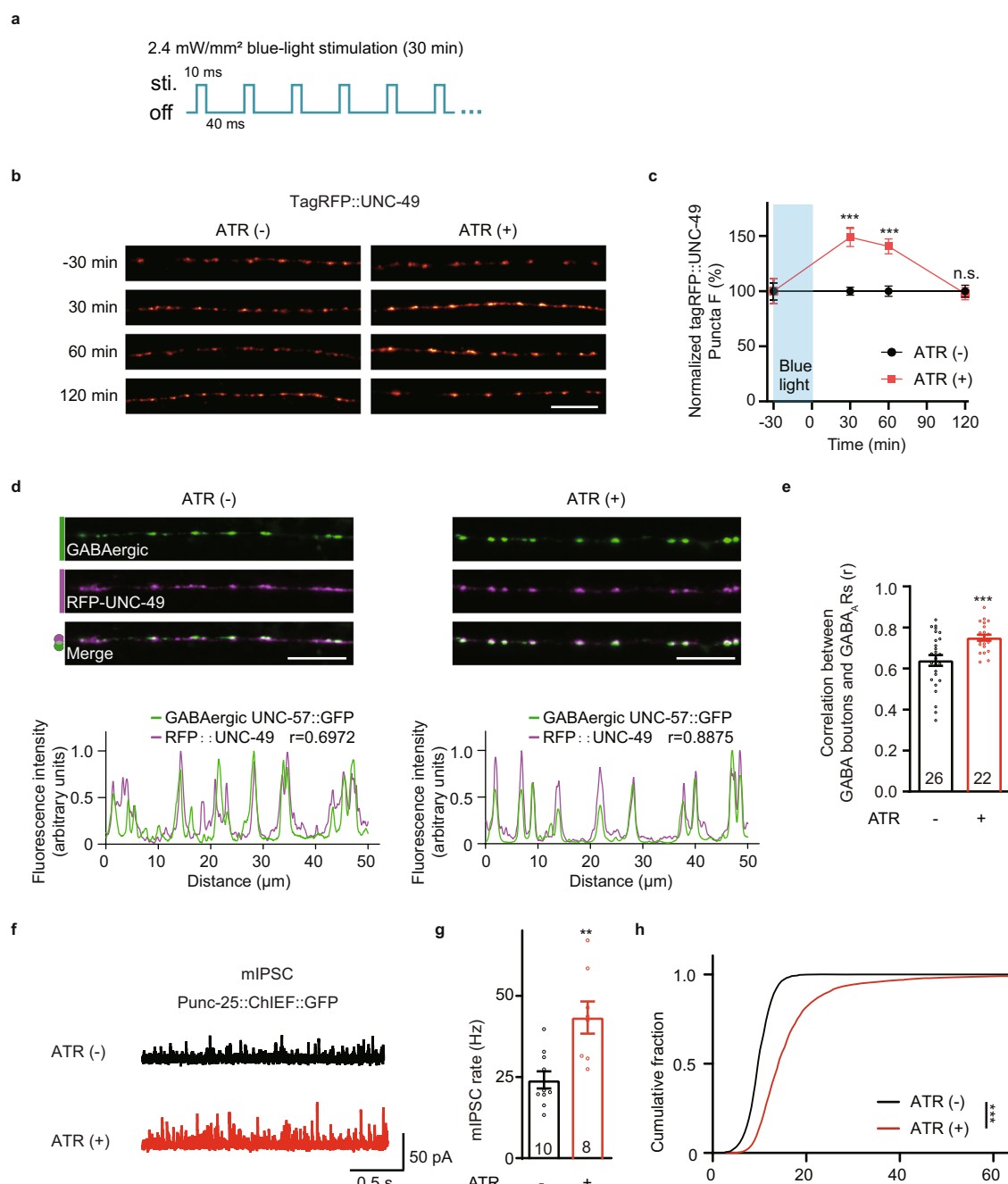

**Fig. 8 | Induction of activity-dependent plasticity at GABAergic synapses.**
**a** Schematic illustration of the blue-light stimulation pattern. **b**, **c** The activity-dependent plasticity at GABAergic synapses lasts within two hours. Representative images (**b**, scale bar 10 μm, Pseudo-color: Red Hot) and the normalized tagRFP fluorescence intensity before and after GABAergic motor neuron excitation (**c**, Data are presented as mean values +/− SEM) are shown. n = 11(ATR(−); −30 min); n = 11(ATR(+); −30 min); n = 28 (ATR(−); 30 min); n = 25 (ATR(+); 30 min); n = 22 (ATR(−); 60 min); n = 24 (ATR(+); 60 min); n = 23 (ATR(−); 120 min); n = 23 (ATR(+); 120 min) animals. Two-tailed and unpaired Student's t-test. ***p < 0.001, n.s. not significant. **d**, **e** The colocalization coefficient between GABA_ARs and GABAergic synaptic boutons was increased after excitation of GABAergic motor neurons; Pearson's correlation coefficients between the intensities of GABAergic bouton marker UNC-57 (green, labeled by UNC-57-split GFP under *unc-25* promoter) and

postsynaptic GABA_ARs (magenta) were used to assess the localization of GABA_ARs at inhibitory synapses. Representative images (**d**-top, scale bar, 10 μm), corresponding line scan curve (**d**-bottom), and mean Pearson's correlation coefficients +/− SEM (**e**, Ns represent the number of animals tested) are shown. Two-tailed and unpaired Student's t-test. ***p < 0.001. **f**–**h** Endogenous inhibitory synaptic transmission was assessed by recording mIPSCs from body-wall muscles in transgenic animals expressing a channelrhodopsin variant (ChIEF) in the GABAergic motor neurons (under *unc-25* promoter) after blue-light stimulation with or without all-trans retinal (ATR). Representative mIPSC traces (**f**), mIPSC rates (**g**, Data are presented as mean values +/− SEM, Ns represent the number of animals tested.), and the cumulative fraction of mIPSC amplitude (**h**) are shown. Two-tailed and unpaired Student's t-test for **g** and Kolmogorov–Smirnov test for **h**. **p < 0.01, ***p < 0.001. For **c**, **e**, **g**, **h**, source data are provided as a Source Data file.

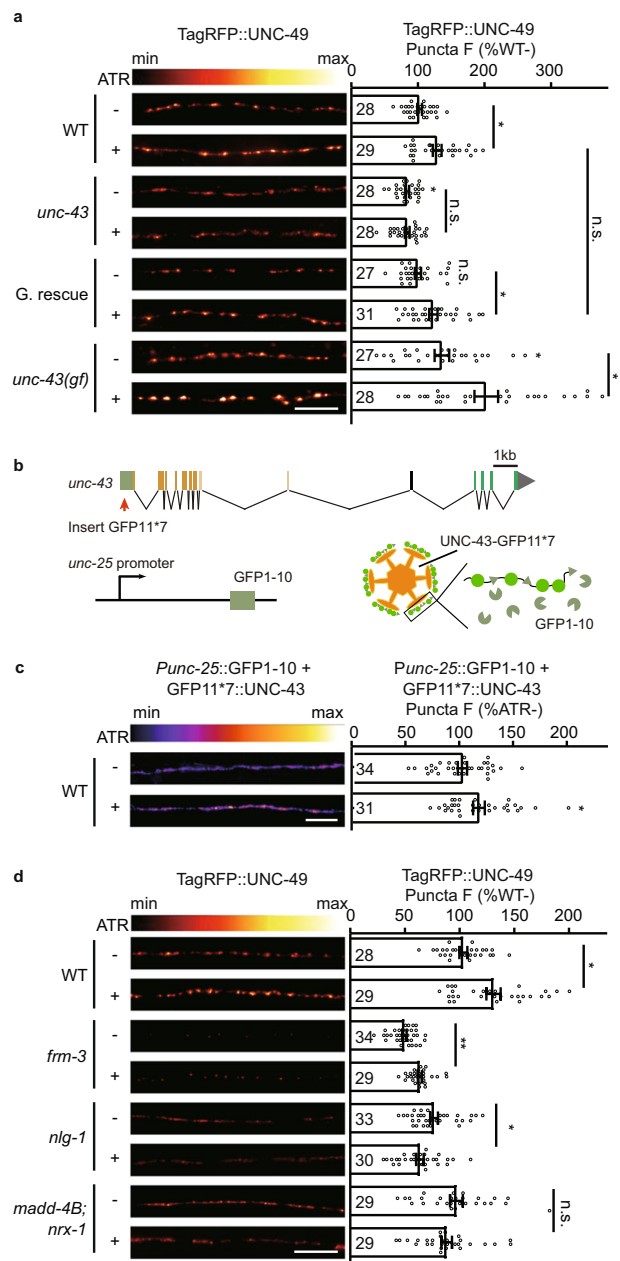

**Fig. 9 | UNC-43/CaMKII is required for activity-dependent plasticity at GABAergic synapses. a** Presynaptic UNC-43 is required for activity-dependent GABAergic plasticity. TagRFP-UNC-49 puncta fluorescence intensity was increased after GABAergic motor neuron excitation by ChIEF activation in wild type and in the *unc-43* gain-of-function mutants, but not in the *unc-43* loss of function mutants, and this was rescued by transgenic expression of UNC-43 in GABAergic motor neurons (G. rescue). Representative images (left, scale bar 10 μm, Pseudo-color: Red Hot) and mean puncta intensities +/− SEM (right, Ns represent the number of animals tested) are shown. Kruskal−Wallis test with two-stage linear step-up procedure of Benjamini, Krieger, and Yekutieli correction for multiple comparisons. *$p < 0.05$, n.s. not significant. **b** Schematic illustration of split GFP experimental design to label endogenous UNC-43 at GABAergic motor neurons. Seven copies of the split GFP11 were inserted into the N-terminal of *unc-43* genomic loci by the CRISPR-Cas9 system. The split GFP1-10 was expressed in GABAergic motor neurons by *unc-25* promoters. **c** The GABAergic presynaptic localization of UNC-43 was increased after GABAergic motor neuron excitation by ChIEF activation. split GFP labels endogenous UNC-43 in GABAergic neurons. Representative images (left, scale bar 10 μm, Pseudo-color: Fire) and mean puncta intensities +/− SEM (right, Ns represent the number of animals tested) are shown. Two-tailed Mann−Whitney test. *$p < 0.05$. **d** MADD-4B, NRX-1α, and NLG-1, but not FRM-3, are required for activity-dependent plasticity at GABAergic synapses. Representative images (left, scale bar 10 μm, Pseudo-color: Red Hot) and mean puncta intensities +/− SEM (right, Ns represent the number of animals tested) are shown. The data for WT is the same as in **a**. Kruskal−Wallis test with two-stage linear step-up procedure of Benjamini, Krieger, and Yekutieli correction for multiple comparisons. *$p < 0.05$, **$p < 0.01$, n.s. not significant. For **a**, **c**, **d**, source data are provided as a Source Data file.

recruited to inhibitory postsynaptic elements upon excitation of pre-synaptic GABAergic motor neurons: it is likely that Ca$^{2+}$ influx after GABAergic motor neuron excitation activates UNC-43, which can trigger more MADD-4B secretion and NRX-1α surface delivery. These molecules can subsequently act as anterograde signals to recruit additional NLG-1 to postsynaptic elements to somehow help stabilize the GABA$_A$Rs. Given that no physical interactions have yet been reported between NLG-1 and GABA$_A$Rs, the nature of this stabilization remains unclear. A recent study proposed that FRM-3 and CASK ortholog LIN-2 mediate the interaction between NLG-1 and GABA$_A$Rs[18]. However, an additive defect in GABA$_A$Rs recruitment and inhibitory synaptic transmission was observed in the *frm-3; nlg-1* double mutant[17]. Further, UNC-43 functions in a single pathway with NLG-1 to recruit GABA$_A$Rs, but not with FRM-3. In summary, the current data does not support the conclusion that NLG-1 recruits GABA$_A$Rs through FRM-3.

## Methods

### Contact for reagent and resource sharing

Further information and requests for resources and reagents should be directed to and fulfilled by the Lead Contact Xia-Jing Tong (tongxj@shanghaitech.edu.cn).

### Animals

All *C. elegans* strains were derived from the wild-type Bristol N2 (Cae-norhabditis Genetics Center) strain. Worms were cultivated at 20 °C on nematode growth medium (NGM) plates seeded with *Escherichia coli* (*E. coli*) under standard conditions. The OP50 strain of *E. coli* was used as a food source for all experiments, except where the HB101 strain was utilized for the electrophysiology study. The well-fed young-adult hermaphrodites were used in all experiments except for coelomocyte imaging experiment in which the adult day-5 animals were used. Transgenic animals were prepared by microinjection, and integrated transgenes were isolated following UV irradiation or by single-copy insertion mediated by miniMos or CRISPR-cas9. A complete list of the strains used in this study can be found in Supplementary data 1.

Molecular information on *unc-43* mutants. The *js125* mutation is a deletion from 10 kb upstream of the transcription initiation site to exon 10[50,58]. The kinase-dead mutant *unc-43(k42r)* was generated by

Recall our finding that the extent of the synaptic GABA$_A$R abundance defects, MADD-4B secretion defect, and NRX-1α motor neuron surface delivery were similar in the *unc-43* null mutant as in worms expressing a kinase-dead mutant *unc-43 (k42r)* (Fig. 1d, Figs. 6 and 7). This suggests that UNC-43's kinase activity is required to promote MADD-4B secretion and NRX-1α surface delivery to support transsynaptic recruitment of GABA$_A$Rs.

## NLG-1- and FRM-3-stabilized GABA$_A$Rs

Previous studies have demonstrated that GABA$_A$Rs are stabilized by distinct synaptic scaffolds at *C. elegans* NMJs: there are NLG-1-stabilized GABA$_A$Rs and FRM-3-stabilized GABA$_A$Rs[15–18]. It has been shown that the NLG-1-stabilized GABA$_A$Rs and FRM-3-stabilized GABA$_A$Rs together mediate inhibitory synaptic transmission[15–18]. However, it remains unclear which type of receptors mediate inhibitory synaptic plasticity. Here, by using the *nlg-1* and *frm-3* mutants, we demonstrate that NLG-1-stabilized GABA$_A$Rs mediate activity-dependent plasticity at inhibitory synapses. Further, our study indicates a plausible explanation for how NLG-1-stabilized GABA$_A$Rs are

CRISPR-cas9, which caused the K42R coding variant. The gain-of-function *unc-43 (n498)* mutation is a point mutation that causes single amino acid substitution (E108K) and makes the kinase partially active even in the absence of $Ca^{2+}$.

Molecular information on *madd-4B* mutants. The *xj0739* mutation is a C insertion in the first exon of *madd-4B*, introducing an early stop codon and generating a 21 amino acids (aa) product.

## Plasmids

The constructs used and created in this study are detailed in Supplementary data 2.

## Primers

The primers used in this study are detailed in Supplementary data 3.

## Reagents, software, and algorithms

The reagents, software, and algorithms used in this study are detailed in Supplementary data 4.

## Genome editing by CRISPR-cas9

The *tagRFP-T-unc-49(xj1024)* knock-in was generated based on the CRISPR-cas9 technique previously described[72]. The tagRFP-T with 3x GS linker was inserted between Gln23 and Asp24, and an extra Gln was added before Asp24 to guarantee the UNC-49 signal peptide coding sequence integrity. The sgRNAs were designed on CRISPR (http://crispor.tefor.net/)[73]. The single-stranded repair template was generated by overlap extension PCR (purified and heated at 95 °C for 5 min). To remove off-target mutations, the transgenic worms were outcrossed with N2 and used for further studies.

Similarly, the *unc-43(k42r)* allele was also generated by CRISPR-cas9. A 99 bp single-stranded repair template was synthesized and purified before injection.

The *unc-43(xj764[LoxP::unc-43p::unc-43::LoxP])* allele was generated for *unc-43* conditional knockout. The first LoxP with EcoRI restriction enzyme cutting site was inserted upstream of *unc-43* promoter (3 kb before exon 1 of UNC-43 transcripts *K11E8.1a* to *K11E8.1 l*), and the second LoxP with EcoRI was inserted into the first intron. Under the action of Cre recombinase driven by the tissue-specific promoter, about 3800 bp genomic DNA containing the promoter and 59 bp coding sequence of UNC-43 transcripts (K11E8.1a to K11E8.1l) including start codon was tissue-specifically removed. For the transcript K11E8.1r, the deletion of the 59 bp coding sequence causes a frameshift and an early stop codon. *myo-3 and unc-25* promoters were cloned into pFX_NLS::Cre vector[74]. The conditional knockout animals were verified by PCR and sequencing.

The *unc-57::GFP11\*7(xj1544)* and *GFP11\*7-unc-43(xj1455)* allele was generated for tissue-specific labeling UNC-57 and UNC-43. *unc-57::GFP11\*7(xj1544)*, seven copies of the GFP11 coding sequence were inserted at the C-terminus of the *unc-57* genomic locus. Briefly, GGGS (linker)-GFP11\*7 coding sequence was inserted before the stop codon of *unc-57* isoforms (T04D1.3a to T04D1.3d). For *GFP11\*7-unc-43(xj1455)*, seven copies of the GFP11 coding sequence were inserted at the N-terminus of the *unc-43* genomic locus. Briefly, GGGS (linker)-GFP11\*7-GGGS (linker) coding sequence was inserted after the start codon of *unc-43* isoforms (K11E8.1b to K11E8.1l). In parallel, the GFP 1-10 fragment was constitutively expressed in the GABAergic motor neurons. As a result, the endogenous UNC-57 and UNC-43 localized in the GABAergic neurons can be visualized.

## Generation of single-copy insertion alleles by miniMos

N2 worms were injected with 10 ng/μL of PCFJ910 plasmid-of-interest containing the promoter and open reading frame (contains a Neomycin resistance gene), 50 ng/μL of pCFJ601 (Mos1 transposase), 10 ng/μL of pGH8 (Prab-3::mCherry), 2.5 ng/μL of pCFJ90 (Pmyo-2::mCherry), and 5 ng/μL of pCFJ104 (Pmyo-3::mCherry). Animals were grown under 25 °C after injection. Neomycin (G418) was added to plates 24 h after injection at a final concentration of 1.5 μg/μL. Homozygous animals with the desired insertion were verified by PCR and sequencing.

The *xjSi0009* allele encodes GFP-tagged MADD-4B driven by GABAergic motor neuron-specific *unc-25* promoter. The GFP sequence was inserted after Phe711 of the MADD-4B coding sequence.

The *xjSi0016* allele encodes dual-tagged NRX-1α driven by the *unc-25* promoter. The pHluorin sequence was inserted between Ile29 and Ile30 of NRX-1α, and the wrmScarlet was inserted after Val1540.

## Microscopy

For UNC-49, UNC-29, UNC-57, UNC-43, FRM-3, and NLG-1 puncta fluorescence imaging, images were captured using a 100x objective (NA = 1.4) on an Olympus microscope (BX53). For ACR-16, MADD-4B, and NRX-1α puncta fluorescence imaging, images were captured using a Nikon 60×1.4 NA objective on a Nikon spinning-disk confocal system (Yokogawa CSU-W1). Young-adult worms were immobilized with 30 μg/μl 2,3-Butanedione monoxime (Sigma). The maximum intensity of dorsal cord projections of Z-series stacks was obtained by Metamorph software (Molecular Devices). Line scans were analyzed in Igor Pro (WaveMetrics) using a custom script[75]. The mean fluorescence intensities of reference FluoSphere microspheres (0.5 μm, ThermoFisher Scientific) were measured during each experiment that controlled for changes in illumination intensities. Automatic image analysis was performed as in previous reports[76]. Briefly, four image parameters were defined: (1) Peak ($F_{peak}$): Absolute peak fluorescence, is the averaged ratio of peak fluorescence to the bead standard of the cord analyzed; (2) Cord ($F_{cord}$): absolute axon fluorescence, which is the ratio of baseline fluorescence to the bead standard; (3) Puncta F: peak-to-cord magnitude, 100X ($F_{peak} - F_{cord}$)/$F_{cord}$; (4) Density: is the number of peaks found per 10 μm of cord analyzed. All fluorescence values are normalized to wild-type controls to facilitate comparison. To assess the synaptic accumulation of fluorescent proteins, we used the Puncta F: peak-to-cord magnitude.

For colocalization assay, images were captured using a 100x objective (NA = 1.4) on an Olympus microscope (BX53). The maximum intensity of dorsal cord projections of Z-series stacks was obtained by ImageJ. The background was removed using the Subtract Background plugin of Image J (rolling ball radius 50 pixels). The fluorescence intensity along the cord was evaluated with the Plot Profile plugin (line width 20 pixels). For each channel, the values were normalized to the value of maximal intensity. To assess the correlations between GABAergic boutons and GABA$_A$Rs, the mean of Pearson's coefficients between the distribution of fluorescence intensity in each channel were compared in Figs. 2d and 8e.

For coelomocytes imaging, images were captured using a Nikon 60 × 1.4 NA objective on a Nikon spinning-disk confocal system (Yokogawa CSU-W1). 5-day adult worms were immobilized with 30 μg/μl 2,3-Butanedione monoxime. All fluorescence values are normalized to wild-type controls to facilitate comparison. To assess fluorescence intensities of coelomocytes, region of interests (ROIs) were traced for the coelomocytes and a background area outside of the animal in ImageJ. Maximum intensities of these ROIs were exported, and fluorescence intensity was corrected for background intensity.

## Super-resolution microscopy

Super-resolution images were captured with a Nikon 60 × 1.4 NA Objective on a Nikon spinning-disk confocal system (Yokogawa CSU-W1 SoRa) based on SoRa mode. Live young adult animals were anesthetized with 30 μg/μl 2,3-Butanedione monoxime (Sigma) and the regions of dorsal cords were excited by a 561 nm laser (50% power, 400 ms exposure time). The maximum intensity of dorsal cord projections of Z-series stacks was obtained by ImageJ. The number of

animals with diffusing or punctate GABA$_A$Rs was assessed and analyzed by Chi-square test.

## Aldicarb assay

The aldicarb assay was performed as previously described[77]. Briefly, 1 mM aldicarb was added to the NGM plate. More than 20 animals at the young-adult stage were picked on each plate. The paralyzed animals were counted every 10 min. At least three double-blind replicates were performed for each genotype.

## Electrophysiology

Electrophysiology was conducted on dissected *C. elegans* as previously described[17]. Worms were superfused in the extracellular solution (127 mM NaCl, 5 mM KCl, 26 mM NaHCO$_3$, 1.25 mM NaH$_2$PO$_4$, 20 mM glucose, 1 mM CaCl$_2$, and 4 mM MgCl$_2$, bubbled with 5% CO$_2$, 95% O$_2$) at 22 °C. Whole-cell recordings were carried out with the internal solution (105 mM CH$_3$O$_3$SCs, 10 mM CsCl, 15 mM CsF, 4 mM MgCl$_2$, 5 mM EGTA, 0.25 mM CaCl$_2$, 10 mM HEPES, and 4 mM Na$_2$ATP). The solution was adjusted to pH 7.2 using CsOH) at 0 mV for mIPSCs.

For GABA-activated current recordings, 0.2 or 0.5 mM GABA was pressure ejected for 1.0 s onto body muscles of adult worms.

## Activity-dependent plasticity assay

To prepare the standard agar plates for optogenetic activation of GABAergic motor neurons, 1.6 mM of all-trans retinal (ATR, 100 mM dissolved in ethanol) or ethanol (control) was mixed with OP50 *E. coli* culture and spotted on 3.5 mm NGM plates. Plates were allowed to dry for 24 h before use. L4 transgenic worms with channelrhodopsin variant CHIEF expressed in the GABAergic motor neurons were transferred to ATR plates and grown overnight. Young adult worms were received pulsed blue-light excitation (460 nm wavelength, 2.4 mW/mm$^2$ power) at 20 Hz for 30 min, and subjected to GABA$_A$Rs puncta fluorescence analysis or electrophysiology.

## Statistical analysis

All data were presented as mean ± SEM (standard error of the mean) with a scatter plot. Data were analyzed with Prism 8.0 (v8.0.2, GraphPad Software, Inc.). The Kolmogorov–Smirnov test was performed to determine whether the data were normally distributed. For comparison of the two groups, a two-tailed unpaired Student's *t*-test (for the normally distributed data, Figs. 3i, 4c, 8c, e, g and Supplementary Figs. 2b, d, 4, 5c, e, f,) or a two-tailed Mann–Whitney test (for the non-normal data, Fig. 9c and Supplementary Fig. 5b, 9a–b, 11a–b) was performed. For the comparison of more than two and less than six groups, one-way ANOVA with post-hoc Bonferroni's multiple comparison test (for the normally distributed data, Figs. 1c, 6b, c, 7c, d and Supplementary Fig. 6c) or Kruskal–Wallis test with post-hoc Dunn's test (for the non-normal data, Figs. 1d, 2d, 3b, f, 4d, 5c, 7e and Supplementary Fig. 6a) or was performed. For the comparison of more than six groups, one-way ANOVA with two-stage linear step-up procedure of Benjamini, Krieger, and Yekutieli correction for multiple comparisons (for the normally distributed data, Fig. 4b and Supplementary Figs. 3, 6b, 8) or Kruskal–Wallis test with two-stage linear step-up procedure of Benjamini, Krieger, and Yekutieli correction for multiple comparisons (for the non-normal data, Figs. 4a, 5a, b, 9a, d) was performed. For the comparison of cumulative fractions, a Kolmogorov–Smirnov test (two group, Fig. 8h) or a Kruskal–Wallis tests with post-hoc Dunn's test (more than two group, Figs. 3c, g) were performed. For Supplementary Fig. 7b, Chi-square tests were performed. For the comparison of curves, two-way ANOVA was performed (Supplementary Fig. 1). *P*-value in all figures is donated as n.s. >0.05, * <0.05, ** <0.005, *** <0.001 (expect Fig. 5a, b and Supplementary Fig. 3, which marked with letters and data corresponding to scatter plots labeled with different letters are significantly different).

## Reporting summary

Further information on research design is available in the Nature Portfolio Reporting Summary linked to this article.

## Data availability

The full raw data that support the results of this study are available upon request. The source data underlying the quantification of Figs. 1c, d, 2d, 3b, c, f, g, 3i, 4a–d, 5a–c, 6b, c, 7c–e, 8c, e, g, h, 9a, c, d and Supplementary Figs. 1, 2b, d, 3, 4, 5b, c, e, f, 6a–c, 7b, 8, 9a, b, 11a, b are provided as a Source data file. Source data are provided with this paper.

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

## Acknowledgements

We thank the C. elegans Genetics Stock Center, National BioResource Project (NBRP), Dr. Joshua Kaplan, Dr.Jean-Louis Bessereau, Dr. Yan Zou, and Dr. Yingchuan B. Qi for sharing strains and reagents. We also thank the Molecular Imaging Core Facility (MICF) at the School of Life Science and Technology, ShanghaiTech University for help in imaging. This work was supported by the STI2030-Major Projects (2021ZD0202500 to X.-J.T.), the National Natural Science Foundation of China (32170963 to X.-J.T.), the Science and Technology Commission of Shanghai Municipality (21ZR1481000 to X.-J.T., 19JC1414100 to X.-J.T.), and the Major International (Regional) Joint Research Project (32020103007 to S.G.).

## Author contributions

Y.H., H.L., X.-T.Z, Y.W., W.-X.Z., K.-Y.Q., L.L., and M.-X.C. designed, performed, and analyzed the experiments. Y.H., X.-T.Z., W.-X.Z., K.-Y.Q., and M.-X.C. performed the aldicarb experiments, fluorescent imaging, and strain construction. H.L., Y.W., and L.L. performed electrophysiological recordings. S.G., Z.H., and X.-J.T. supervised the experimental design and data interpretation. Y.H. and X.-J.T. wrote the manuscript. All authors discussed the results and commented on the manuscript.

## Competing interests

The authors declare no competing interests.
