## [Peer Review File · Nature Communications]

UNC-43/CaMKII-triggered anterograde signals recruit GABAARs to mediate inhibitory synaptic transmission and plasticity at *C. elegans* NMJsREVIEWERS' COMMENTS:

Reviewer #1 (Remarks to the Author):

Anterograde signaling from presynaptic neurons play very important roles in the clustering of postsynaptic receptors. A great deal is known about how presynaptic signals regulate postsynaptic receptor clustering at cholinergic synapses of neuromuscular junctions, and at glutamatergic synapses in the brain. However the presynaptic signal(s) and the signaling pathway mediating the clustering of postsynaptic GABA receptors are less well understood. This study shows that UNC-43/CaMKII in GABAergic motor neurons recruit postsynaptic GABAARs transsynaptically through a signaling pathway that also includes the presynaptic proteins MADD-4B/punctin, and NRX-1 α /neurexin, and the postsynaptic protein NLG-1/neuroigin. Furthermore, this study shows that UNC-43/CaMKII is required for activity-dependent plasticity at GABAergic synapses. The conclusions are mostly supported by rigorous genetic and cell biological experiments. The findings are potentially of broad interest. However, there are a few issues that may deserve the authors' considerations.

Major points:

1. It was concluded that UNC-43 functions in a single pathway with NLG-1 to recruit GABAARs because adding *unc-43(lf)* to *frm-3(lf)* but not *nlg-1(lf)* caused a further reduction in the TagRFP::*UNC-49* signal compared with the corresponding *frm-3* or *nlg-1* single mutant (Figure 4A). However, the difference between *frm-3* and *unc-43;frm-3* was small albeit statistically significant. Because *unc-43(lf)* reduced the TagRFP::*UNC-49* signal by more than 30%, the difference between *frm-3* and *unc-43;frm-3* is expected to be much larger than what was observed if UNC-43 works in a FRM-3-independent pathway. Therefore, it would be helpful to confirm the conclusion by testing whether the augmenting effect of *unc-43* gain-of-function(gf) on the TagRFP::*UNC-49* signal (Figure 1D) can be eliminated by *nlg-1(lf)* but not *frm-3(lf)* in *unc-43(n498);nlg-1* and *unc-43(n498);frm-3* double mutants.
2. One important conclusion of this study was that UNC-43 is required for activity-dependent plasticity at GABAergic synapses. This conclusion was based on differences in GABAAR puncta signal and aldicarb sensitivity between worms with and without all-trans retinal (ATR) treatment. However, the difference in aldicarb sensitivity between the two groups was small. Because the aldicarb assay often fails to reflect synaptic transmission at the neuromuscular junction, it would be helpful to confirm the conclusion by comparing properties of spontaneous GABAergic postsynaptic currents between worms with and without the ATR treatment.
3. Several fusion proteins were used in this study, such as C-terminal GFP-tagged NLG-1, and C-terminal GFP-tagged MADD-4B. The validity of conclusions reached with them depends to some extent on whether the fusion proteins are functional. The authors should test whether they are functional by performing mutant rescue experiments.

Minor points:

1. Page 6, lines 15-16. "UNC-43 is expressed in both pre- and postsynapses at excitatory and inhibitory synapses." Please provide references for this sentence.
2. Page 6, lines 16-19. "To examine where UNC-43 functions to stabilize the postsynaptic GABAARs, we restored UNC-43 expression in GABAergic motor neurons (under the *unc-25* promoter), in cholinergic motor neurons (under the *unc-17* promoter), or in body-wall muscles (under the *myo-3* promoter)". It is better to provide a simple description about the synaptic relationships among cholinergic motor neurons, GABAergic motor neurons, and body-wall muscle cells at the beginning of this paragraph because many readers outside of the *C. elegans* community do not know that cholinergic motor neurons are presynaptic to GABAergic motor neurons in *C. elegans*.
3. Page 6, lines 22-24. "The partial rescue of synaptic abundance of GABAARs may be caused by

the involvement of multiple UNC-43 isoforms." Please describe how many isoforms UNC-43 has, and why you chose this isoform for your rescue experiments.

4. Page 7, lines 13-15. "We found that the colocalization coefficient in the *unc-43* mutant was significantly decreased compared with the wild type (0.361 ± 0.035 for wild type vs. 0.0923 ± 0.057 for *unc-43*) (Fig. 2A-B)." Please explain what the colocalization coefficient means. Would a value of 1 indicate perfect colocalization of two proteins? Please try to offer an interpretation why the colocalization coefficient is much lower than 1 in the wild type, and describe the range of typical colocalization coefficient values between postsynaptic receptors and presynaptic markers at *C. elegans* neuromuscular junctions.

Reviewer #2 (Remarks to the Author):

In this paper, Hao et al describe a role for CaMKII/*unc-43* in controlling the abundance of postsynaptic GABA(A) receptors/*unc-49*. In *unc-43* mutants the levels of postsynaptic *unc-49* are reduced, mIPSC rate is reduced, and the ability of presynaptic optogenetic stimulation to recruit postsynaptic *unc-49* is reduced. Using genetic analysis, the authors link *unc-43* to a previously described pathway consisting of presynaptic Neurexin and Punctin, which control postsynaptic Neuroligin and *unc-49*. They provide evidence that *unc-43* is required for efficient secretion of Punctin and surface delivery of Neurexin.

The strengths of the manuscript include the experimental system, which the authors nicely leverage to observe the endogenous localization of *unc-49*, *unc-43* and cholinergic receptors. The genetic analysis is comprehensive and includes the generation of cell-specific conditional alleles. Furthermore, clever experiments testing the secretion of Punctin and surface delivery of neurexin point to a potential site of *unc-43* action (but see comment below).

The main weakness is the lack of a mechanism for how *unc-43* works. This would be important both for the novelty aspect (since the Neurexin/Madd-4B, NLG pathway was previously described) and for ruling out that the effects that the authors observe reflect defective synapse development/maintenance rather than a direct effect on Neurexin and Madd-4 secretion. This concern is particularly relevant given the known developmental roles of *unc-43* in worms and CaMKII functions in regulating dense-core vesicles (which can strongly affect synapse properties). Specific comments (in order of importance):

- Given the concerns mentioned above, the over expression of *unc-57* is not sufficient as a control that presynapse development is not impaired.
- The authors claim that *unc-43* acts through NLG based on an analysis of fluorescence intensity. However, looking at the images, the phenotypes seem different: *unc-49* is much more diffuse in neuroligin mutants, whereas it is still punctate in *unc-43* mutants.
- Related to the previous point: it would be good to measure additional synaptic organization parameters other than fluorescence intensity (for example, number of puncta per micron).
- For the measurements of puncta intensity, the description doesn't make it clear if what the authors describe as "peak" is the intensity of the brightest punctum or the combined intensity of all puncta.
- Is the secretion of Neurexin and Madd-4B increased during stimulation? The authors have the tools to show this.
- The genetic model would be strengthened by showing suppression of *unc-43* (*gf*) by neurexin and *madd-4* mutants.
- For the Phluorin assay it would be better to normalize each data point by RFP rather than show that on average RFP levels do not change.
- The plasticity the authors observe is a somewhat puzzling. Both presynaptic loss of function (*unc-25* mutants) and presynaptic gain of function (photostimulation) lead to increased GABA receptor levels. This should be addressed, at least in the discussion.
- The first line in the Discussion is misleading and overclaims: the trans-synaptic mechanism was described in previous studies by other groups, not in this study.
- The same *unc-43* animal is shown twice (Figures 4C and 5B).

Reviewer #3 (Remarks to the Author):

The manuscript Hao et al. 2022 describes a presynaptic role for the CaMKII UNC-43 in GABAAR recruitment to postsynapses in *C. elegans*. The authors show that UNC-43 functions presynaptically from GABAergic motor neurons to promote the accumulation of endogenously-tagged postsynaptic GABAARs in muscle, and that this role of UNC-43 is dependent on its kinase activity. Using conditional tissue-specific knockouts, they show that a reduction in postsynaptic mIPSC amplitude in *unc-43* mutants is recapitulated in a presynaptic GABA motor neuron-specific *unc-43* knockout, but not a postsynaptic muscle-specific *unc-43* knockout, suggesting that UNC-43 acts from presynaptic GABAergic neurons to affect postsynaptic GABAAR clustering and mediate its role in inhibitory transmission. The authors go on to show that UNC-43 is acting from presynaptic GABAergic motor neurons to regulate postsynaptic Neuroligin (NLG-1) abundance. They provide evidence that this occurs via UNC-43 regulation of secretion of the signaling molecule MADD-4B and surface localization of neurexin (NRX-1). Finally, using channelrhodopsin-based excitation of GABAergic neurons, the authors observed activity-dependent increases in GABAAR fluorescence intensity that was dependent on UNC-43 expression in GABAergic neurons, and found that UNC-43 itself becomes more concentrated at presynapses upon stimulation. The authors also linked NLG-1, MADD-4B, and NRX-1 to this activity-dependent plasticity of GABAAR postsynaptic abundance. This work reveals the very interesting presynaptic role of CaMKII/UNC-43 in regulating GABAAR recruitment to postsynapses.

This manuscript requires only minor revisions which are discussed chronologically:

Figure 1: The partial rescue of GABA-R synaptic abundance by GABAergic motor neuron expression of UNC-43 is not satisfactorily explained. First, it would be helpful to the reader to go into greater detail about the rescuing construct being used in this study – which isoform(s) of *unc-43* does it encode? Can a minigene that incorporates a portion of the genomic region and therefore encodes multiple isoforms be used for rescue experiments? Alternatively, the authors should check for any synergistic effects of rescue in cholinergic and GABAergic neurons, ie- by using a pan-neuronal promoter. A requirement for UNC-43 in cholinergic neurons might depend on it also being present in GABAergic neurons, and thus only be apparent when rescued in both.

Figure 2: It looks like both the pre- and post-synaptic markers are affected by the *unc-43* mutation, however the text describes the UNC-43 mutation as affecting localization only of GABAARs. The text should be adjusted to reflect that more general observation of a change in the registration of the pre- and post-synaptic compartments, rather than just a mislocalization of the post-synaptic side. Additionally, the TagRFP::*UNC-49* does not appear as affected by the *unc-43* mutant in Figure 2A as it does in Figure 1C,D.

Figure 3: Panel D provides a description of the insertion sites of LoxP in the *unc-43* gene, however it is unclear which isoform of *unc-43* this map refers to, as there are multiple different start sites of *unc-43*.

Figure 5: Datasets of various genotypes are used repeatedly in multiple panels of Figure 5. While this does not necessarily represent an issue, further clarification of how the statistics were performed is necessary. In particular, [emphasis:] multiple comparisons should be performed for all the data together first, prior to separating into individual graphs. Otherwise the number of comparisons is artificially low, which could lead to the perception of significant differences. Furthermore, we do not see the need to display the data in so many different graphs: A-B and separately C-F could be combined into larger panels. This would allow comparisons to be made between genotypes that are currently separated into different panels. For example, further discussion on why the double *madd-4B; nrx-1* mutant is more severe than the single *unc-43* mutant would be helpful. Finally, the representative image in 5F of the *madd-4B;unc-43;nrx-1* triple mutant looks much more severe than the representative image of *unc-43; nrx-1* double mutant in 5E.

Reviewer #4 (Remarks to the Author):

Hao and co-authors investigate signaling by postsynaptic GABA-A receptors at the *C. elegans* NMJ. In this model system, inhibitory GABAergic neurons directly inhibit postsynaptic muscle function via activation of postsynaptic GABA-A receptors. Previous published work from several laboratories, including from the senior author, demonstrated that the clustering of postsynaptic GABAR at muscles is dependent on the secretion of MADD-4/Punctin by presynaptic GABAergic motor neurons which interacts with presynaptic NRX-1 and postsynaptic NLG-1 scaffold proteins. These protein interactions localize/stabilize postsynaptic GABA-A receptors on muscle cells.

This manuscript builds on the previously discovered molecular architecture and addresses whether presynaptic signaling by CaMKII regulates the clustering and thus the signaling mediated by postsynaptic GABA-A receptors. The authors primarily rely on *in vivo* imaging of fluorescently tagged proteins and electrophysiology. The authors argue that UNC-43/CaMKII functions solely in presynaptic GABAergic motor neurons to regulate the secretion of MADD-4B and the surface expression of NRX-1. By this mechanism postsynaptic GABA-A receptor localization/stabilization is enhanced. The activity of CaMKII has been demonstrated in many studies to be regulated by electrical activity via changes in intracellular Ca²⁺, and as is well known in mammalian experiments, CaMKII has been demonstrated to have an essential role in cellular models of synaptic plasticity such as long-term potentiation (LTP). Here, the authors argue that GABAergic synapses in *C. elegans* also exhibit activity-dependent plasticity, and that this plasticity is regulated by presynaptic CaMKII that acts via MADD-4/Punctin and NRX-1.

The manuscript uses technically state-of-the-art approaches to investigate receptor localization and function and makes excellent use of available mutants to investigate the neuronal regulation of synaptic signaling mediated by GABA-A receptors. I have three major concerns:

A) The major contribution appears somewhat incremental, that is, the authors primarily demonstrate that CaMKII functions exclusively at presynaptic neurons to regulate postsynaptic inhibitory signaling mediated by GABA-A receptors. UNC-43/CaMKII has many targets and even in *C. elegans* it has already been shown to regulate neurotransmitter release by phosphorylating potassium channels (Liu et al, *J. Neurosci*, 2007). The level of mechanistic analysis here falls significantly short of comparable papers (e.g., Petrini et al., *Nature Comm*, 2014) and does not adequately address how CaMKII modifies signaling mediated by postsynaptic GABA-A receptors. Importantly, the authors do not separate the known effects of CaMKII on release of synaptic vesicles from any possible effects on NRX-1 and MADD-4.

B) The authors argue for an exclusive presynaptic role for CaMKII. The genetic rescue experiments do not fully support this conclusion, nor does an earlier electrophysiological study in *C. elegans*.

C) The plasticity experiments are very preliminary and difficult to interpret. No discussion of relevant work studying the inhibitory LTP in mammals and the role of postsynaptic CaMKII.

Specific Comments:

1. The authors do not provide the necessary background to motivate the study of presynaptic CaMKII with respect to the regulation and plasticity of postsynaptic GABA-A receptors, thus the choice to study CaMKII appears somewhat arbitrary. A rich vertebrate literature argues that postsynaptic CaMKII regulates postsynaptic receptors and inhibitory LTP (e.g., Petrini et al., *Nature Comm*, 2014, which provides a detailed mechanistic analysis; Houston et al., *J. Physiol*, 2009). The authors introduce their study by speculating that transsynaptic recruitment of GABA-A receptors might occur, and if so, CaMKII might have a role in this process. The study of CaMKII appears motivated solely by the importance of this protein for mammalian LTP. The authors do not adequately discuss the vertebrate literature in the introduction nor do they place their findings in the context of a larger literature in the discussion.

2. The major finding of the paper is the reduced fluorescence intensity of a tagRFP-GABA-A receptor fusion protein in CaMKII mutants (Fig. 1). The decrease is modest. The authors argue that the defect can be rescued by selective expression of CaMKII in presynaptic GABAergic

neurons. However, the rescue is only partial, calling into question the interpretation of the results. Indeed, partial rescue is found in other figures as well (e.g., Fig. 3, 4).

The issue of partial rescue is especially important since an earlier paper (Liu et al, J. Neurosci, 2007) investigated the role of CaMKII in GABA-A receptor mediated signaling and suggests that CaMKII has both presynaptic (regulation of synaptic vesicle/neurotransmitter release) and postsynaptic (surface expression of GABA-A receptors) roles. The authors argue that the incomplete rescue might be due to a requirement for multiple CaMKII isoforms. Alternatively, rescue might be dependent on the strength of the promoters. There is no independent measurement of how much CaMKII is expressed in the different cell types. These alternative models can be easily addressed with additional rescue experiments using different isoforms, including cell-specific expression of the CaMKII(gf) variant. Additionally, the authors should provide control experiments demonstrating the percent rescue using the native CaMKII promoter.

3. The authors argue that CaMKII regulates MADD-4 secretion and the presynaptic delivery of the transmembrane protein NRX-1. With respect to secretion of MADD-4 (Fig. 6), controls are required to address whether the observed secretion defect is due to a general defect in protein secretion by presynaptic neurons.

With respect to surface expression of NRX-1 (Fig. 7), by eye the example images do not appear different – if anything, the overall fluorescence appears decreased in mutants compare to wild type. The interpretation of these experiments is difficult since essential controls are lacking (e.g., changing intracellular pH). Thus, the observed change in NRX-1 expression could simply reflect unc-43-mediated changes in intracellular pH.

4. An earlier paper (Liu et al., J. Neurosci., 2007)), demonstrated that GABA-A receptor-mediated currents are reduced in unc-43/CaMKII mutants. In contrast, no defect is found in the present manuscript. The authors argue that this is because Liu et al. use 0.2 mM GABA and this paper uses 0.5 mM GABA. Both concentrations are well above the EC50 for GABA-A receptors. The authors should resolve this important discrepancy by testing currents evoked by application of 0.2 mM GABA.

5. In Fig. 8 the authors present evidence for activity dependent changes in GABA-A receptors (possible plasticity). This data is difficult to assess because of the lack of foundation/experimental details. By what definition is this plasticity? What are the properties of this plasticity? How does CaMKII contribute? Is the plasticity occluded by gain-of-function CaMKII?

RESPONSE TO REVIEWERS' COMMENTS

Responses to Reviewer #1: pages 1 to 8

Responses to Reviewer #2: pages 9 to 20

Responses to Reviewer #3: pages 21 to 27

Responses to Reviewer #4: pages 28 to 41

Responses to Reviewer #1:

Anterograde signaling from presynaptic neurons play very important roles in the clustering of postsynaptic receptors. A great deal is known about how presynaptic signals regulate postsynaptic receptor clustering at cholinergic synapses of neuromuscular junctions, and at glutamatergic synapses in the brain. However the presynaptic signal(s) and the signaling pathway mediating the clustering of postsynaptic GABA receptors are less well understood. This study shows that UNC-43/CaMKII in GABAergic motor neurons recruit postsynaptic GABAARs transsynaptically through a signaling pathway that also includes the presynaptic proteins MADD-4B/punctin, and NRX-1 α /neurexin, and the postsynaptic protein NLG-1/neuroigin. Furthermore, this study shows that UNC-43/CaMKII is required for activity-dependent plasticity at GABAergic synapses. The conclusions are mostly supported by rigorous genetic and cell biological experiments. The findings are potentially of broad interest. However, there are a few issues that may deserve the authors' considerations.

Major points:

1. It was concluded that UNC-43 functions in a single pathway with NLG-1 to recruit GABAARs because adding *unc-43(lf)* to *frm-3(lf)* but not *nlg-1(lf)* caused a further reduction in the TagRFP::UNC-49 signal compared with the corresponding *frm-3* or *nlg-1* single mutant (Figure 4A). However, the difference between *frm-3* and *unc-43;frm-3* was small albeit statistically significant. Because *unc-43(lf)* reduced the TagRFP::UNC-49 signal by more than 30%, the difference between *frm-3* and *unc-43;frm-3* is expected to be much larger than what was observed if UNC-43 works in a FRM-3-independent pathway. Therefore, it would be helpful to confirm the conclusion by testing whether the augmenting effect of *unc-43* gain-of-function(*gf*) on the TagRFP::UNC-49 signal (Figure 1D) can be eliminated by *nlg-1(lf)* but not *frm-3(lf)* in *unc-43(n498);nlg-1* and *unc-43(n498);frm-3* double mutants.

Response:

We appreciate the reviewer's great suggestion, and analyzed GABA_ARs recruitment in the *unc-43 (gf)*; *nlg-1* and *unc-43 (gf)*; *frm-3* mutants as suggested. *unc-43 (gf)* mutation causes an increase in TagRFP::UNC-49 puncta fluorescence

intensity in the *frm-3* mutants, but not in the *nlg-1* mutants, supporting that UNC-43 functions in a single pathway with NLG-1 to recruit GABA_ARs.

Figure 4. UNC-43/CaMKII acts through a single pathway with NLG-1/neuroigin to recruit GABA_ARs. (B) The tagRFP-UNC-49 puncta fluorescence intensities in the *unc-43(gf); nlg-1* double mutants were comparable to the *nlg-1* single mutants. Representative images (left, scale bar 10 μm) and mean puncta intensities (right) in wild type and mutants are shown. Ns represent the number of animals tested, *** p<0.001, n.s. not significant. One-way ANOVA with two-stage linear step-up procedure of Benjamini, Krieger, and Yekutieli correction for multiple comparisons.

We have added this data in Figure 4B, and revised our manuscript on page 10, lines 18-21, which reads as follows:

“Further, the increase of GABA_AR fluorescence intensity by *unc-43(n498)* gain-of-function mutation was eliminated in the *nlg-1* mutant, but not in the *frm-3* mutant (Fig. 4B), lending more support to the conclusion that UNC-43 functions in a single pathway with NLG-1 to recruit GABA_ARs.”

2. One important conclusion of this study was that UNC-43 is required for activity-dependent plasticity at GABAergic synapses. This conclusion was based on differences in GABA_AR puncta signal and aldicarb sensitivity between worms with and without all-trans retinal (ATR) treatment. However, the difference in aldicarb sensitivity between the two groups was small. Because the aldicarb assay often fails to reflect synaptic transmission at the neuromuscular junction, it would be helpful to confirm the conclusion by comparing properties of spontaneous GABAergic postsynaptic currents between worms with and without the ATR treatment.

Response:

This is also a good suggestion. According to the reviewer’s suggestion, we recorded the mIPSC in animals with GABAergic motor neurons excited. Compared with the control groups without all-trans retinal (ATR) treatment, those animals presented significantly increased mIPSC frequency and amplitude, supporting that the inhibitory synaptic transmission was also increased.

Figure 8. Induction of activity-dependent plasticity at GABAergic synapses. (F-H) Endogenous inhibitory synaptic transmission was assessed by recording mIPSCs from body-wall muscles in transgenic animals expressing a channelrhodopsin variant (ChIEF) in the GABAergic motor neurons (under *unc-25* promoter) after blue-light stimulation with or without all-trans retinal (ATR). Representative mIPSC traces (F), mIPSC rates (G), and the cumulative fraction of mIPSC amplitude (H) are shown. *Ns* represent the number of animals tested. ** $p < 0.01$, *** $p < 0.001$. unpaired Student's t-test for G, Kolmogorov-Smirnov test for H.

We have added this data in Figure 8F-H, and revised our manuscript on page 14, lines 6-12, which reads as follows:

“To study whether the increased GABA_AR recruitment upon GABAergic motor neuron excitation potentiates inhibitory synaptic transmission, we patch-clamped body-wall muscles and recorded mIPSCs. Both the mIPSCs frequency and amplitude were significantly increased in those animals with GABAergic motor neuron excited compared to controls without all-trans retinal (ATR) (Fig. 8F-H), supporting that the inhibitory synaptic transmission was also increased.”

3. Several fusion proteins were used in this study, such as C-terminal GFP-tagged NLG-1, and C-terminal GFP-tagged MADD-4B. The validity of conclusions reached with them depends to some extent on whether the fusion proteins are functional. The authors should test whether they are functional by performing mutant rescue experiments.

Response:

We used imaging analysis to validate the normal function of tagged fusion proteins including MADD-4B-GFP, NLG-1-GFP, and FRM-3-GFP. The transgene that expresses MADD-4B-GFP in the GABAergic motor neuron (under *unc-25* promoter) is able to fully rescue the defect of GABA_AR puncta fluorescence intensities in the *madd-4B* mutants. A transgene that expresses NLG-1-GFP in the muscle cells (under *myo-3* promoter) rescues the decreased GABA_AR puncta fluorescence intensities in the *nlg-1* mutants. Further, the transgene that expresses FRM-3-GFP in the muscle cells (under *myo-3* promoter) partially rescues the GABA_AR puncta fluorescence intensities in the *frm-3* mutants. The partial rescue of GABA_AR recruitment was also observed in *frm-3* mutants expressing non-tagged full-length FRM-3 (PMID: 26575289). Collectively, the data support the conclusion that the fusion proteins, including MADD-4B-GFP, NLG-1-GFP, and FRM-3-GFP, function normally.

Supplementary Figure 8. Fusion proteins are able to rescue the GABA_ARs recruitment defects in their corresponding mutants. TagRFP-UNC-49 fluorescence in dorsal nerve cords in wild type and mutants is shown. Representative images (left, scale bar 10 μm) and mean puncta intensity (right) are shown. Pseudo-color: Red Hot. Ns represent the number of animals tested. one-way ANOVA with two-stage linear step-up procedure of Benjamini, Krieger, and Yekutieli correction for multiple comparisons.

We have added this data in Supplementary Figure 8, and revised our manuscript on page 10, lines 25-27, page 11, lines 3-4, and page 12, lines 12-16, which reads as follows:

“The GABA_ARs recruitment defects of *nlg-1* mutants were rescued by the transgene expression of the C-terminal GFP-tagged NLG-1 in the muscle cells (Supplementary Fig. 8).”

“It is worth noting that the postsynaptic localization of FRM-3 was unaltered in the *unc-43* mutants (Fig. 4C, Supplementary Fig. 8).”

“To study MADD-4B secretion, we expressed a C-terminal GFP-tagged MADD-4B fusion in GABAergic neurons (under the *unc-25* promoter), and the MADD-4B-GFP fusion protein is able to rescue the GABA_ARs recruitment defects in the *madd-4b* mutants (Supplementary Fig. 8).”

Minor points:

Text 1. Page 6, lines 15-16. “UNC-43 is expressed in both pre- and postsynapses at excitatory and inhibitory synapses.” Please provide references for this sentence.

Response:

We have added the citations after this statement.

Text 2. Page 6, lines 16-19. “To examine where UNC-43 functions to stabilize the postsynaptic GABA_ARs, we restored UNC-43 expression in GABAergic motor neurons (under the *unc-25* promoter), in cholinergic motor neurons (under the

unc-17 promoter), or in body-wall muscles (under the *myo-3* promoter)". It is better to provide a simple description about the synaptic relationships among cholinergic motor neurons, GABAergic motor neurons, and body-wall muscle cells at the beginning of this paragraph because many readers outside of the *C. elegans* community do not know that cholinergic motor neurons are presynaptic to GABAergic motor neurons in *C. elegans*.

Response:

We appreciate the reviewer's good suggestion. We have included the description in the manuscript on page 6, lines 20 to 22, which reads as follows:

"At *C. elegans* NMJs, the excitatory motor neurons innervate and contract muscles, and also synapses onto the inhibitory motor neurons to relax the contralateral muscles and generate the sinusoidal movement ⁵²."

3. Page 6, lines 22-24. "The partial rescue of synaptic abundance of GABAARs may be caused by the involvement of multiple UNC-43 isoforms." Please describe how many isoforms UNC-43 has, and why you chose this isoform for your rescue experiments.

Response:

There are 18 UNC-43 isoforms according to WormBase, and both isoforms UNC-43D and UNC-43G have been shown to be able to mediate UNC-43's synaptic roles (PMID: 10647013; PMID: 10647014; PMID: 17898212; PMID: 23663262). Here we used both UNC-43D and UNC-43G isoforms to perform the rescue experiments, and used the UNC-43D isoform for imaging analysis.

We have revised the manuscript on pages 6-7, which reads as follows:

"To examine where UNC-43 functions to stabilize the postsynaptic GABA_ARs, we restored UNC-43 expression in GABAergic motor neurons (under the *unc-25* promoter), in cholinergic motor neurons (under the *unc-17* promoter), or in body-wall muscles (under the *myo-3* promoter) (all in the *unc-43* mutant background). We found that the decreased synaptic abundance of GABA_ARs in the *unc-43* mutant was partially rescued by expressing either UNC-43D or UNC-43G isoform ⁵⁹⁻⁶¹ in GABAergic motor neurons; no rescue was observed with the cholinergic neuron or body-wall muscle groups (Fig. 1C and Supplementary Fig. 3). All of the data supports that UNC-43 functions in the GABAergic motor neurons to stabilize postsynaptic GABA_ARs."

4. Page 7, lines 13-15. "We found that the colocalization coefficient in the *unc-43* mutant was significantly decreased compared with the wild type (0.361 ± 0.035 for wild type vs. 0.0923 ± 0.057 for *unc-43*) (Fig. 2A-B)." Please explain what the colocalization coefficient means. Would a value of 1 indicate perfect colocalization of two proteins? Please try to offer an interpretation why the colocalization coefficient is much lower than 1 in the wild type, and describe the range of typical colocalization coefficient values between postsynaptic receptors and presynaptic markers at *C. elegans* neuromuscular junctions.

Response:

The colocalization coefficient is to characterize the degree of overlap between two channels in the microscopy image. The value can range from 1 (perfect positive correlation) to -1 (perfect negative correlation). According to the published literature, the colocalization coefficient values between postsynaptic receptors and presynaptic markers range from 0.6 to 0.8 (PMID: 26028575).

In the originally submitted figure, we overexpressed the GFP-RAB-3 fusion protein in the GABAergic motor neurons, and checked its colocalization coefficient with tagRFP-GABA_ARs. We appreciate the reviewer's reminder, which helps us to realize that overexpression of fusion protein may cause a mislocalization, and underestimate the colocalization coefficient. To solve this problem, we labeled the endogenous UNC-57/Endophilin with GFP in the GABAergic synapses by split GFP complementary system, and we measured its correlation coefficient with tagRFP-GABA_ARs. The results showed that in the wild type, their correlate coefficient is 0.7506. Consistent with the originally submitted results, the colocalization coefficient was significantly decreased in the *unc-43* mutants, and can be rescued by restoring UNC-43 expression in the GABAergic motor neurons.

We have replaced Figure 2 with the new data, and revised the manuscript on pages 7-8, which reads as follows:

"To further investigate whether the remaining GABA_ARs in the *unc-43* mutant are correctly localized at GABAergic synapses, we labeled the GABAergic synapses by fusing the endogenous UNC-57/Endophilin in GABAergic motor neurons with GFP in the tagRFP-tagged GABA_ARs strain using a split GFP complementation system^{63,64} (Fig. 2A), and calculated the colocalization coefficient between UNC-57 and GABA_ARs (Fig. 2B). We found that the colocalization coefficient in the *unc-43* mutant was significantly decreased compared with the wild type (0.7770 ± 0.0143 for wild type vs. 0.4848 ± 0.0316 for *unc-43*) (Fig. 2C-D). Further, GABA_AR mislocalization could be rescued by restoring UNC-43 expression in the GABAergic motor neurons (Fig. 2C-D), indicating that presynaptic UNC-43 is required for the GABAergic synaptic localization of GABA_ARs."

Figure 2. Presynaptic UNC-43/CaMKII is required for the GABAergic synaptic localization of GABA_ARs. (A-B) Schematic illustration of labeling UNC-57 at the GABAergic motor neurons by split GFP complementary system. Seven copies of the split GFP11 were inserted into the C-terminal of *unc-57* genomic loci by CRISPR-Cas9 system. The split GFP1-10 was expressed in GABAergic motor neurons by *unc-25* promoters. (C-D) The colocalization coefficient between GABA_ARs and GABAergic synaptic boutons was decreased in the *unc-43* mutants; this was rescued by transgenic expression of UNC-43 in GABAergic motor neurons (G. rescue). Pearson's correlation coefficients between the intensities of GABAergic bouton marker UNC-57 (green, labeled by UNC-57-split GFP under *unc-25* promoter) and postsynaptic GABA_ARs (red) were used to assess the localization of GABA_ARs at inhibitory synapses. Representative images (C-top, scale bar, 10 μm), corresponding line scan curve (C-bottom), and mean Pearson's correlation coefficients (D) are shown. In D, Ns represent the number of animals tested. *** p<0.001. Kruskal-Wallis test with post-hoc Dunn's test.

We would like to take this opportunity to express our gratitude for the helpful guidance from the reviewer about how to improve our study and manuscript. Many thanks.

Responses to Reviewer #2:

In this paper, Hao et al describe a role for CaMKII/unc-43 in controlling the abundance of post-synaptic GABA(A) receptors/unc-49. In *unc-43* mutants the levels of postsynaptic unc-49 are reduced, mIPSC rate is reduced, and the ability of presynaptic optogenetic stimulation to recruit postsynaptic unc-49 is reduced. Using genetic analysis, the authors link *unc-43* to a previously described pathway consisting of presynaptic Neurexin and Punctin, which control postsynaptic Neuroligin and unc-49. They provide evidence that *unc-43* is required for efficient secretion of Punctin and surface delivery of Neurexin.

The strengths of the manuscript include the experimental system, which the authors nicely leverage to observe the endogenous localization of *unc-49*, *unc-43* and cholinergic receptors. The genetic analysis is comprehensive and includes the generation of cell-specific conditional alleles. Furthermore, clever experiments testing the secretion of Punctin and surface delivery of neurexin point to a potential site of *unc-43* action (but see comment below).

The main weakness is the lack of a mechanism for how *unc-43* works. This would be important both for the novelty aspect (since the Neurexin/Madd-4B, NLG pathway was previously described) and for ruling out that the effects that the authors observe reflect defective synapse development/maintenance rather than a direct effect on Neurexin and Madd-4 secretion. This concern is particularly relevant given the known developmental roles of *unc-43* in worms and CaMKII functions in regulating dense-core vesicles (which can strongly affect synapse properties). Specific comments (in order of importance):

Given the concerns mentioned above, the over expression of *unc-57* is not sufficient as a control that presynapse development is not impaired.

Response:

We appreciate the reviewer's rigorous thoughts and constructive suggestions. To better study the presynapse development and rule out the possibility that the GABA_ARs recruitment defects in the *unc-43* mutants were secondary effects of abnormal presynapse development/maintenance, we did the following experiments:

(1) Study the endogenous UNC-57/Endophilin localization at GABAergic synapses in the *unc-43* mutants

We labeled the endogenous UNC-57/Endophilin with GFP in the GABAergic synapses by split GFP complementary system, and analyzed the puncta fluorescence intensity and density of the UNC-57-GFP fusion protein. We observed no significant differences in UNC-57-GFP puncta fluorescence intensities and densities between wild-type and *unc-43* mutant animals.

Figure 2. (A) Schematic illustration of labeling UNC-57 at the GABAergic motor neurons by split GFP complementary system. Seven copies of the split GFP11 were inserted into the C-terminal of *unc-57* genomic loci by CRISPR-Cas9 system. The split GFP1-10 was expressed in GABAergic motor neurons by *unc-25* promoters.

Supplementary Figure 5. GABAergic synapse structure is unaltered by deletion of *unc-43*. (A-C) The puncta fluorescence intensities and densities—marked by the GABAergic UNC-57::split GFP (under *unc-25* promoter)—are unaltered in the *unc-43* mutants. Representative images (A), mean puncta intensities (B), and puncta densities (C) are shown. Pseudo-color: Fire. Ns represent the number of animals tested. n.s. not significant. Mann Whitney test for B and unpaired Student’s t-test for C.

(2) Study the endogenous UNC-2/CaV2 localization at GABAergic synapses in the *unc-43* mutants

Further, we labeled the endogenous UNC-2 in the GABAergic synapses with GFP by split GFP complementary system. We found that the puncta fluorescence intensities and densities of UNC-2-GFP fusion proteins were unaltered in the *unc-43* mutants.

Supplementary Figure 5. GABAergic synapse structure is unaltered by deletion of *unc-43*. (D-F) Split GFP complementary system to label the endogenous UNC-2 at the GABAergic synapses. The puncta fluorescence intensities and densities—marked by the GABAergic UNC-2::split GFP (under *unc-47* promoter)—are unaltered in the

unc-43 mutants. Representative images (D), mean puncta intensities (E), and puncta density (F) are shown. Pseudo-color: Fire. Ns represent the number of animals tested. n.s. not significant. Unpaired Student's t-test for E-F.

(3) Study whether synaptic vesicle release is required for transsynaptic GABA_AR recruitment

In our originally submitted manuscript, we ruled out the possibility that the decreased postsynaptic GABA_ARs in the *unc-43* mutants result from diminished presynaptic GABA transmission, as both a previous report and our data have demonstrated that the synaptic abundance of GABA_AR was not decreased in the *unc-25* mutants (which lack GABA biogenesis) and *unc-13* mutants (which lack synaptic vesicle release) (Supplementary Figure 4, it's Supplementary Figure 6 in the current version of manuscript). In the revised manuscript, we also measured the GABA_ARs puncta fluorescence in the *snb-1* mutant, another mutant that presents severe synaptic vesicle release defects. We did not observe defects in GABA_ARs recruitment in the *snb-1* mutants. Those data suggest that synaptic vesicle release is not required to recruit postsynaptic GABA_ARs, and support the conclusion that the decreased postsynaptic GABA_ARs in the *unc-43* mutants does not result from diminished presynaptic GABA transmission.

(4) Study whether dense-core vesicle release and neuropeptide secretion are required for transsynaptic GABA_AR recruitment

Further, we ruled out the possibility that the GABA_ARs recruitment defects in the *unc-43* mutants result from abnormal presynaptic dense-core vesicles release or neuropeptide secretion, as GABA_ARs puncta fluorescence intensities are not decreased in the *unc-31* mutants (lack of dense-core vesicle release) and *egl-3* mutants (lack of neuropeptide maturation).

A**B****C**
Supplementary Figure 6. Lack of synaptic vesicle or dense-core vesicle release does not cause a decrease in GABA_AR abundance at synapses. TagRFP-UNC-49 puncta fluorescence intensities in *unc-25* (A), *slo-1*, *unc-13*, *snb-1* (B), *unc-31* and *egl-3* (C) mutants. Representative images (scale bar 10 μ m) and mean puncta intensity are shown. Pseudo-color: Red Hot. Ns represent the number of animals tested. ***, $p < 0.001$. Unpaired Student's t-test for A, one-way ANOVA with post-hoc Bonferroni's multiple comparison test for B and C.

(5) Study whether UNC-43/CaMKII regulate general presynaptic protein secretion

To study whether UNC-43/CaMKII regulates general presynaptic protein secretion, we expressed a constitutive secretion GFP from the GABAergic motor neurons (under *unc-25* promoter), and measured the secreted GFP that

endocytosis by the scavenger cell coelomocytes. We observed no significant differences in GFP fluorescence intensities in coelomocytes between wild type and *unc-43* mutants, suggesting that UNC-43/CaMKII does not regulate general protein secretion in the GABAergic motor neurons, pointing toward that UNC-43 specifically regulates MADD-4B secretion and the surface delivery of NRX-1 α .

B

Supplementary Figure 9. The endocytosis of coelomocytes and general presynaptic protein secretion are not affected by *unc-43* mutation. (B) The constitutive secretion of GFP from GABAergic motor neuron terminals was not affected in the *unc-43* mutants. Secretion of GFP was measured by analyzing GFP fluorescence intensities in the coelomocytes. Scale bar 5 μ m. The representative images (left panel) and the mean fluorescence intensities are shown (right panel). n.s. not significant. Mann Whitney test.

In summary, these data suggest that i) the GABAergic synapse structure was unaltered in the *unc-43* mutants supporting that the decreased postsynaptic GABA_ARs in the *unc-43* mutant is not the secondary result of synaptic structural defects; ii) the GABA_ARs recruitment defect in the *unc-43* mutants does not result from an abnormal synaptic vesicle or dense-core vesicle release; iii) Finally, we showed that UNC-43 does not regulate general presynaptic protein secretion, and it has specificity in regulating protein secretion/surface delivery. Our new data provide more underlying mechanisms and support the conclusion that UNC-43 directly regulates MADD-4B secretion and NRX-1 surface delivery. We hope our newly added data addresses the reviewer's concern.

We have added the data in Figure 2, Supplementary Fig. 5, Supplementary Fig. 6, and Supplementary Fig. 9.

The authors claim that *unc-43* acts through NLG based on an analysis of fluorescence intensity. However, looking at the images, the phenotypes seem different: *unc-49* is much more diffuse in neurologin mutants, whereas it is still punctate in *unc-43* mutants.

- Related to the previous point: it would be good to measure additional synaptic organization parameters other than fluorescence intensity (for example, number of puncta per micron).

Response:

We appreciate the reviewer's constructive suggestion. To better study the distribution of GABA_ARs, we analyzed GABA_ARs clustering by super-resolution microscopy (Nikon W1 SoRa mode). Briefly, GABA_ARs form clusters along the nerve cord in the wild-type animals, and their signals were diffusing in the *nlg-1* mutants and *unc-43* mutants, but not in the *frm-3* mutants. The percentage of animals with normal GABA_ARs clustering was significantly decreased in both *nlg-1* and *unc-43* mutants, but not in the *frm-3* mutants, supporting that UNC-43 acts in a single pathway with NLG-1 to recruit GABA_ARs.

A

B

Supplementary Figure 7. GABA_ARs clustering is impaired in the *unc-43* mutants. Quantification of GABA_ARs clustering in the *unc-43*, *frm-3*, and *nlg-1* mutants by Super-resolution microscopy studies (Sora mode). Representative images (scale bar 10 μ m) and the percentage of animals with normal GABA_AR clusters are shown. Pseudo-color: Red Hot. Ns represent the number of animals tested. ***, $p < 0.001$, n.s. not significant. Chi-square tests.

We have added the results in Supplementary Figure 7, and revised the manuscript on page 10, lines 15-17, which reads as follows:

“Besides, Super-resolution microscopy studies by Sora mode showed that the GABA_ARs clustering was significantly decreased in both the *unc-43* mutants and the *nlg-1* mutants, but not in the *frm-3* mutants (Supplementary Fig. 7).”

- For the measurements of puncta intensity, the description doesn't make it clear if what the authors describe as “peak” is the intensity of the brightest punctum or the combined intensity of all puncta.

Response:

We now realize that our description of the method did not make it clear enough to follow. Here, each “Peak” value is the averaged peak fluorescence of the cord analyzed normalized to the bead standard.

We appreciate the reviewer for raising this question, and we revised our manuscript on page 21, lines 22-24, which reads as follows:

“Automatic image analysis was performed as in previous reports ⁷⁴. Briefly, four image parameters were defined: (1) Peak (Fpeak): Absolute peak fluorescence, is the averaged ratio of peak fluorescence to the bead standard of the cord analyzed;”

- Is the secretion of Neurexin and MADD-4B increased during stimulation? The authors have the tools to show this.

Response:

(1) We analyzed the surface delivery of NRX-1 α during the induction of activity-dependent plasticity at the GABAergic synapses, and observed a dramatic increase of the pHluorin fluorescence intensities, while the wrmScarlet fluorescence intensities were unaltered, indicating that the GABAergic motor neuron surface delivery of NRX-1 α , but not its expression or axonal trafficking, was increased during the GABAergic motor neurons excited.

(2) We also measured MADD-4B secretion upon GABAergic motor neurons excitation. Note that the coelomocyte assay is not able to detect transient changes; thus, we conducted experiments in which we added the pH-sensitive pHluorin fusion tag at the N-terminus of MADD-4B and monitored the secreted MADD-4B that was retained at synaptic membranes by binding to other synaptic membrane proteins, such as NLG-1. We observed a significant increase in the pHluorin fluorescence intensities upon blue light-stimulated GABAergic neuron excitation, supporting the conclusion that MADD-4B secretion is potentiated.

A

B

Supplementary Figure 11. The secretion of MADD-4B and surface delivery of NRX-1 α were increased after GABAergic motor neuron excitation by ChIEF activation. (A) The pHluorin fusion tag was fused at the N-terminus of MADD-4B, and its fluorescent signal indicates the secreted MADD-4B that was retained at synaptic membranes. pHluorin-MADD-4B puncta fluorescence intensity in the dorsal nerve cords was increased in transgenic animals expressing a channelrhodopsin variant (ChIEF) in the GABAergic motor neurons (under *unc-25* promoter) after blue-light stimulation with all-trans retinal (ATR). (B) The surface localization of NRX-1 α is increased in transgenic animals expressing ChIEF in the GABAergic motor neurons after blue-light stimulation with all-trans retinal (ATR). Representative images (upper), the averaged fluorescence intensity of pHluorin (lower, left), wrmScarlet (lower, middle), and the fluorescence intensity of pHluorin normalized to wrmScarlet (lower, right) are shown. Scale bar 10 μ m. Ns represent the number of animals tested. * $p < 0.05$, ** $p < 0.01$, n.s. not significant. Mann Whitney test.

We have added the results in Supplementary Figure 11, and revised the manuscript on page 15, lines 19-22, which reads as follows:

“Further, the secretion of MADD-4B and the surface delivery of NRX-1 α were both increased upon GABAergic neuron excitation (Supplementary Fig.11), supporting the conclusion that MADD-4 and NRX-1 α also participate in the activity-dependent plasticity at GABAergic synapses, and they are very likely to work in the single pathway with UNC-43”

- The genetic model would be strengthened by showing suppression of *unc-43* (*gf*) by neurexin and *madd-4* mutants.

Response:

We appreciate the reviewer’s constructive suggestion, and analyzed GABA_ARs recruitment in the *madd-4B; unc-43* (*gf*); *nrx-1* triple mutants as recommended. *unc-43* (*gf*) mutation causes an increase of TagRFP::UNC-49 puncta fluorescence in the wild type, but not in the *madd-4B; nrx-1* double mutants, supporting that UNC-43 recruits postsynaptic GABA_ARs requiring both MADD-4B and NRX-1 α .

Figure 5. UNC-43/CaMKII’s recruitment of GABA_ARs requires both MADD-4B and NRX-1 α . (B-C) Both MADD-4B and NRX-1 α are required for UNC-43’s recruitment of GABA_ARs. TagRFP-UNC-49 fluorescence in dorsal nerve cords in wild type and mutants is shown. Representative images (left, scale bar 10 μ m) and mean puncta intensity (right) are shown. In C, Pseudo-color: Red Hot. Ns represent the number of animals tested. In C, ** $p < 0.01$, *** $p < 0.001$, n.s. not significant. Kruskal-Wallis test with post-hoc Dunn’s test.

We have added this data in Figure 5C, and revised our manuscript on page 12, lines 4-7, which reads as follows:

“Further, *unc-43(n498)* gain-of-function mutation is not able to increase the GABA_AR puncta fluorescence intensity in the *madd-4b; nrx-1* double mutants (Fig. 5C), indicating that UNC-43 recruits postsynaptic GABA_ARs requiring both MADD-4B and NRX-1 α .”

Text - For the Phluorin assay it would be better to normalize each data point by RFP rather than show that on average RFP levels do not change.

Response:

According to the reviewer’s suggestion, we revised Figure 7 by normalizing each data point by RFP.

Figure 7. UNC-43/CaMKII promotes NRX-1 α /neurexin GABAergic motor neuron surface localization. (B-E) The surface localization of NRX-1 α is decreased in the *unc-43* mutants. pFluorin and wrmScarlet dual-labeled NRX-1 α was expressed in GABAergic neurons. pFluorin-NRX-1 α puncta fluorescence in the dorsal nerve cords was decreased in the *unc-43* mutants (C). NRX-1 α -wrmScarlet puncta fluorescence in the dorsal nerve cords was unaltered in the *unc-43* mutants (D). The averaged fluorescence intensity of pFluorin normalized to wrmScarlet fluorescence intensity in the *unc-43* mutants (E). Representative images and mean dorsal cord puncta intensity are shown. Scale bar for B 10 μ m. In C and D, Ns represent the number of animals tested. *** $p < 0.001$, n.s. not significant. One-way ANOVA with post-hoc Bonferroni's multiple comparison test for C-D, Kruskal-Wallis test with post-hoc Dunn's test for E.

Text - The plasticity the authors observe is a somewhat puzzling. Both presynaptic loss of function (*unc-25* mutants) and presynaptic gain of function (photostimulation) lead to increased GABA receptor levels. This should be addressed, at least in the discussion.

Response:

According to the reviewer's suggestion, we have revised the manuscript on pages 10, lines 2-3, which reads as follows: "The increase of GABA $_A$ Rs puncta fluorescence in the *unc-25* mutants may be caused by synaptic homeostasis."

Text - The first line in the Discussion is misleading and overclaims: the trans-synaptic mechanism was described in previous studies by other groups, not in this study.

Response:

We appreciate the reviewer for clarifying. We have revised the manuscript on page 15, lines 25-26, which reads as follows:

"In this study, we revealed how UNC-43/CaMKII participates in transsynaptically recruiting GABA $_A$ Rs at NMJs."

Text - The same *unc-43* animal is shown twice (Figures 4C and 5B).

Response:

We thank the reviewer for pointing out this question. All of the imaging analyses in Figure 4C (Figure 4D in the current version) and Figure 5A were performed parallelly, and separated into different panels. According to the suggestion, we replaced the representative imaging of *unc-43* mutants in Figure 5A, and included the description in Figure 5 legends reads as follows:

“The data for *unc-43* is the same as in Figure 4C.”

We would like to take this opportunity to express our gratitude for the helpful guidance from the reviewer about how to improve our study and manuscript. Many thanks.

Responses to Reviewer #3:

The manuscript Hao et al. 2022 describes a presynaptic role for the CaMKII UNC-43 in GABAAR recruitment to postsynapses in *C. elegans*. The authors show that UNC-43 functions presynaptically from GABAergic motor neurons to promote the accumulation of endogenously-tagged postsynaptic GABAARs in muscle, and that this role of UNC-43 is dependent on its kinase activity. Using conditional tissue-specific knockouts, they show that a reduction in postsynaptic mIPSC amplitude in *unc-43* mutants is recapitulated in a presynaptic GABA motor neuron-specific *unc-43* knockout, but not a postsynaptic muscle-specific *unc-43* knockout, suggesting that UNC-43 acts from presynaptic GABAergic neurons to affect postsynaptic GABAAR clustering and mediate its role in inhibitory transmission. The authors go on to show that UNC-43 is acting from presynaptic GABAergic motor neurons to regulate postsynaptic Neuroligin (NLG-1) abundance. They provide evidence that this occurs via UNC-43 regulation of secretion of the signaling molecule MADD-4B and surface localization of neuroligin (NRX-1). Finally, using channelrhodopsin-based excitation of GABAergic neurons, the authors observed activity-dependent increases in GABAAR fluorescence intensity that was dependent on UNC-43 expression in GABAergic neurons, and found that UNC-43 itself becomes more concentrated at presynapses upon stimulation. The authors also linked NLG-1, MADD-4B, and NRX-1 to this activity-dependent plasticity of GABAAR postsynaptic abundance.

This work reveals the very interesting presynaptic role of CaMKII/UNC-43 in regulating GABAAR recruitment to postsynapses.

This manuscript requires only minor revisions which are discussed chronologically:

Figure 1: The partial rescue of GABA-R synaptic abundance by GABAergic motor neuron expression of UNC-43 is not satisfactorily explained. First, it would be helpful to the reader to go into greater detail about the rescuing construct being used in this study – which isoform(s) of *unc-43* does it encode? Can a minigene that incorporates a portion of the genomic region and therefore encodes multiple isoforms be used for rescue experiments? Alternatively, the authors should check for any synergistic effects of rescue in cholinergic and GABAergic neurons, ie- by using a pan-neuronal promoter. A requirement for UNC-43 in cholinergic neurons might depend on it also being present in GABAergic neurons, and thus only be apparent when rescued in both.

Response:

We appreciate the reviewer's constructive suggestion, and did the following experiments to test these ideas:

(1) We found that both UNC-43D and UNC-43G isoforms are able to partially rescue the GABA_ARs recruitment defects in the *unc-43* mutants.

According to previous research, both UNC-43D and UNC-43G have been shown to be able to mediate UNC-43's synaptic roles (PMID: 10647013; PMID: 10647014; PMID: 17898212; PMID: 23663262). Here we used the UNC-43D isoform to perform the rescuing experiments. In the revision process, we also tested the rescuing effect of UNC-43G isoform. Similar to UNC-43D, expressing UNC-43G in the GABAergic motor neurons (under *unc-25* promoter) partially rescued the GABA_ARs recruitment defects.

(2) Restoring UNC-43D expression under its own promoter fully rescues the GABA_ARs recruitment defects in the *unc-43* mutants.

According to the reviewer's suggestion, we drove UNC-43D isoform expression under its endogenous promoter in the *unc-43* mutants, and the GABA_ARs puncta fluorescence defects in the *unc-43* mutants can be fully rescued, which rules out the possibility that the partial rescue of the synaptic abundance of GABA_ARs may be caused by the involvement of multiple UNC-43 isoforms, pointing toward the possibility that UNC-43 from other cells also participate in recruiting GABA_ARs.

(3) Restoring UNC-43D expression in pan-neurons or in both GABAergic and cholinergic motor neurons fully rescues the GABA_ARs recruitment defects in the *unc-43* mutants.

Further, we restored UNC-43 (UNC-43D isoform) expression in pan-neurons (under *rab-3* promoter), in both GABAergic and cholinergic motor neurons (*unc-25* promoter+*unc-17* promoter), and in both GABAergic motor neurons and muscle cells (*unc-25* promoter+*myo-3* promoter) in the *unc-43* mutants. We found that the GABA_ARs recruitment defects in the *unc-43* mutants can be fully rescued by expressing UNC-43 in pan-neurons or in both GABAergic and cholinergic motor neurons, but not with both GABAergic motor neurons and muscle cells groups. The data suggest that both the cholinergic and GABAergic UNC-43 are involved GABA_ARs recruitment, ruling out the involvement of postsynaptic UNC-43.

(4) The cholinergic UNC-43 also participates in recruiting GABA_ARs at inhibitory synapses, and it relies on the GABAergic UNC-43.

Recalling the results in Figure 1C and Figure 3A-C that restoring UNC-43 expression in the cholinergic neurons alone has no effects on rescuing both GABA_ARs recruitment defects nor mIPSC defects in *unc-43* mutants. Collectively, all of the data support the conclusion that the cholinergic UNC-43 also participates in recruiting GABA_ARs at inhibitory synapses with some unknown mechanisms, and it relies on the GABAergic UNC-43.

In summary, our data support the conclusion that the UNC-43 in the GABAergic motor neurons transsynaptically recruits GABA_ARs. Although the UNC-43 in the cholinergic motor neurons is able to additively recruit GABA_ARs, it relies on the UNC-43 in the GABAergic motor neurons. We appreciate the reviewer's constructive suggestion, which leads us to new discoveries. We hope we are able to elucidate how UNC-43 in the cholinergic neurons facilitates GABA_AR recruitment in our next paper.

Supplementary Figure 3. Rescue of GABA_ARs recruitment defect in *unc-43* mutants. The decreased TagRFP-UNC-49 puncta fluorescence intensity was rescued by transgenic expression of UNC-43 D or G isoform in GABAergic motor neuron (under *unc-25* promoter), endogenous expressing cells (under *unc-43* promoter), pan neuron (under *rab-3* promoter) both GABAergic motor neurons (under *unc-25* promoter) and cholinergic motor neurons (under *unc-17* promoter), and both in GABAergic motor neurons (under *unc-25* promoter) and body-wall muscles (under *myo-3* promoter). Representative images (left, scale bar 10 μm) and mean puncta intensities (right) are shown. Pseudo-color: Red Hot. Ns represent the number of animals tested. Data corresponding to scatter plot labeled with different letters are significantly different ($p < 0.05$). One-way ANOVA with two-stage linear step-up procedure of Benjamini, Krieger, and Yekutieli correction for multiple comparisons.

We added the data in Supplementary Figure 3, and revised the manuscript on page 7, lines 4-14, which reads as follows:

" Since we observed a partial rescue of GABA_ARs recruitment defects in *unc-43* mutants by expressing UNC-43 in the GABAergic motor neurons, therefore, we restored UNC-43 (UNC-43D isoform) expression in pan-neurons (under *rab-3* or *unc-43* promoter), in both GABAergic and cholinergic motor neurons (under *unc-25* promoter+*unc-17* promoter), and in both GABAergic motor neurons and muscle cells (under *unc-25* promoter+*myo-3* promoter) in the *unc-43* mutants, and we found that the GABA_ARs puncta fluorescence defects in the *unc-43* mutants can be fully rescued by expressing UNC-43 in pan-neurons or in both GABAergic and cholinergic motor neurons, but not with both GABAergic motor neurons and muscle cells groups (Supplementary Fig. 3). These data suggest that the cholinergic UNC-43 is also involved in GABA_ARs recruitment with unknown mechanisms, and it depends on the GABAergic UNC-43."

Text Figure 2: It looks like both the pre- and post-synaptic markers are affected by the *unc-43* mutation, however the text describes the UNC-43 mutation as affecting localization only of GABAARs. The text should be adjusted to reflect that more general observation of a change in the registration of the pre- and post-synaptic compartments, rather than just a mislocalization of the post-synaptic side. Additionally, the TagRFP::UNC-49 does not appear as affected by the *unc-43* mutant in Figure 2A as it does in Figure 1C,D.

Response:

(1) Currently, our data support the conclusion that *unc-43* mutation did not affect the presynaptic structure at GABAergic synapses. In the revision process, we performed two additional experiments to further support this conclusion.

First, we labeled the endogenous UNC-57/Endophilin with GFP in the GABAergic synapses by split GFP complementary system, and analyzed the puncta fluorescence intensity and density of the UNC-57-GFP fusion protein. We observed no significant decrease in puncta fluorescence intensities and densities in the *unc-43* mutants compared to wild-type animals.

Second, we also labeled the endogenous UNC-2 with GFP in the GABAergic synapses by split GFP complementary system, and the puncta fluorescence intensities and densities of UNC-2-GFP fusion proteins were unaltered in the *unc-43* mutants.

Supplementary Figure 5. GABAergic synapse structure is unaltered by deletion of *unc-43*. (A-C) The puncta fluorescence intensities and densities—marked by the GABAergic UNC-57::split GFP (under *unc-25* promoter)—are unaltered in the *unc-43* mutants. Representative images (A), mean puncta intensities (B), and puncta density (C) are shown. (D-F) Split GFP complementary system to label the endogenous UNC-2 at the GABAergic synapses. The puncta fluorescence intensities and densities—marked by the GABAergic UNC-2::split GFP (under *unc-47* promoter)—are unaltered in the *unc-43* mutants. Representative images (D), mean puncta intensities (E), and puncta density (F) are shown. Pseudo-color: Fire. Ns represent the number of animals tested. n.s. not significant. Mann Whitney test for B and unpaired Student’s t-test for C and E-F.

We added the data in Supplementary Figure 5, and revised the manuscript on pages 9, lines 14-22, which reads as follows:

“To rule out the possibility that the decreased postsynaptic GABA_ARs in the *unc-43* mutant is the secondary result of synaptic structural defects, we measured the puncta fluorescence and intensity of endogenous UNC-57/Endophilin in GABAergic motor neurons in the *unc-43* mutant. We observed no significant decrease in puncta fluorescence intensities and densities between wild type and *unc-43* mutants (Supplementary Fig. 5A-C). Further, we also labeled the endogenous UNC-2/CaV2 in the GABAergic motor neurons with GFP by split GFP complementation system⁶⁴. Both

the fluorescence intensity and density of UNC-2-GFP fusion protein were not altered in the *unc-43* mutants compared to the wild type (Supplementary Fig. 5D-F). These results suggested that the GABAergic synapse structure was unaltered.”

(2) Further, according to the reviewer’s suggestion, we applied the same treatment with all of the images in the revised Figure 2, and now they can reflect the intensity decrease in the *unc-43* mutants.

Figure 3: Panel D provides a description of the insertion sites of LoxP in the *unc-43* gene, however it is unclear which isoform of *unc-43* this map refers to, as there are multiple different start sites of *unc-43*.

Response:

According to the reviewer’s suggestion, we revised our method to include all the required information on page 20, lines 2-11, which reads as follows:

“The *unc-43(xj764[LoxP::unc-43p::unc-43::LoxP])* allele was generated for *unc-43* conditional knockout. The first LoxP with EcoRI restriction enzyme cutting site was inserted upstream of *unc-43* promoter (3 kb before exon 1 of UNC-43 transcripts K11E8.1a to K11E8.1l), and the second LoxP with EcoRI was inserted into the first intron. Under the action of Cre recombinase driven by the tissue-specific promoter, about 3800 bp genomic DNA containing the promoter and 59 bp coding sequence of UNC-43 transcripts (K11E8.1a to K11E8.1l) including start codon was tissue-specifically removed. For the transcript K11E8.1r, the deletion of the 59 bp coding sequence causes a frameshift and an early stop codon. *myo-3* and *unc-25* promoters were cloned into pFX_NLS::Cre vector ⁷². The conditional knockout animals were verified by PCR and sequencing.”

Text Figure 5: Datasets of various genotypes are used repeatedly in multiple panels of Figure 5. While this does not necessarily represent an issue, further clarification of how the statistics were performed is necessary. In particular, [emphasis:] multiple comparisons should be performed for all the data together first, prior to separating into individual graphs. Otherwise the number of comparisons is artificially low, which could lead to the perception of significant differences. Furthermore, we do not see the need to display the data in so many different graphs: A-B and separately C-F could be combined into larger panels. This would allow comparisons to be made between genotypes that are currently separated into different panels. For example, further discussion on why the double *madd-4B*; *nrx-1* mutant is more severe than the single *unc-43* mutant would be helpful. Finally, the representative image in 5F of the *madd-4B*;*unc-43*;*nrx-1* triple mutant looks much more severe than the representative image of *unc-43*; *nrx-1* double mutant in 5E.

Response:

We appreciate the reviewer’s great suggestion.

(1) We have reorganized the figures in Figure 5 by combing A-B together and C-F together. We also compared all the data by the Kruskal-Wallis test with multiple comparison, and executed a two-stage linear step-up procedure of Benjamini, Krieger, and Yekutieli.

(2) We also have replaced the representative images in Figure 5F (Figure 5B in the current version) for *madd-4B;unc-43;nrx-1* triple mutants and *unc-43; nrx-1* double mutants.

We would like to take this opportunity to express our gratitude for the helpful guidance from the reviewer about how to improve our study and manuscript. Many thanks.

Responses to Reviewer #4:

Hao and co-authors investigate signaling by postsynaptic GABA-A receptors at the *C. elegans* NMJ. In this model system, inhibitory GABAergic neurons directly inhibit postsynaptic muscle function via activation of postsynaptic GABA-A receptors. Previous published work from several laboratories, including from the senior author, demonstrated that the clustering of postsynaptic GABAR at muscles is dependent on the secretion of MADD-4/Punctin by presynaptic GABAergic motor neurons which interacts with presynaptic NRX-1 and postsynaptic NLG-1 scaffold proteins. These protein interactions localize/stabilize postsynaptic GABA-A receptors on muscle cells.

This manuscript builds on the previously discovered molecular architecture and addresses whether presynaptic signaling by CaMKII regulates the clustering and thus the signaling mediated by postsynaptic GABA-A receptors. The authors primarily rely on *in vivo* imaging of fluorescently tagged proteins and electrophysiology. The authors argue that UNC-43/CaMKII functions solely in presynaptic GABAergic motor neurons to regulate the secretion of MADD-4B and the surface expression of NRX-1. By this mechanism postsynaptic GABA-A receptor localization/stabilization is enhanced. The activity of CaMKII has been demonstrated in many studies to be regulated by electrical activity via changes in intracellular Ca²⁺, and as is well known in mammalian experiments, CaMKII has been demonstrated to have an essential role in cellular models of synaptic plasticity such as long-term potentiation (LTP). Here, the authors argue that GABAergic synapses in *C. elegans* also exhibit activity-dependent plasticity, and that this plasticity is regulated by presynaptic CaMKII that acts via MADD-4/Punctin and NRX-1.

The manuscript uses technically state-of-the-art approaches to investigate receptor localization and function and makes excellent use of available mutants to investigate the neuronal regulation of synaptic signaling mediated by GABA-A receptors. I have three major concerns:

A) The major contribution appears somewhat incremental, that is, the authors primarily demonstrate that CaMKII functions exclusively at presynaptic neurons to regulate postsynaptic inhibitory signaling mediated by GABA-A receptors. UNC-43/CaMKII has many targets and even in *C. elegans* it has already been shown to regulate neurotransmitter release by phosphorylating potassium channels (Liu et al, *J. Neurosci*, 2007). The level of mechanistic analysis here falls significantly short of comparable papers (e.g., Petrini et al., *Nature Comm*, 2014) and does not adequately address how CaMKII modifies signaling mediated by postsynaptic GABA-A receptors. Importantly, the authors do not separate the known effects of CaMKII on release of synaptic vesicles from any possible effects on NRX-1 and MADD-4.

Response:

We appreciate the reviewer's constructive suggestions, and performed additional experiments to address the reviewer's concern:

(1) We studied whether synaptic vesicle release is required for transsynaptic GABA_AR recruitment

In our originally submitted manuscript, we ruled out the possibility that the decreased postsynaptic GABA_AR in the *unc-43* mutants result from diminished presynaptic GABA transmission, as both a previous report and our data have demonstrated that the synaptic abundance of GABA_AR was not decreased in the *unc-25* mutants (which lack GABA

biogenesis) and *unc-13* mutants (which lack synaptic vesicle release) (Supplementary Figure 4, it's Supplementary Figure 6 in the current version of manuscript).

In the revised manuscript, we also measured the GABA_ARs puncta fluorescence in the *snb-1* mutant, another mutant that presents severe synaptic vesicle release defects. Consistent with the results in *unc-13* mutants, we did not observe a decrease in GABA_ARs recruitment.

These results support that the decreased GABA_ARs recruitment in the *unc-43* mutants does not result from diminished presynaptic vesicle transmission.

(2) We studied whether dense-core vesicle release or neuropeptide secretion is required for transsynaptic GABA_AR recruitment

Further, we studied GABA_ARs puncta fluorescence intensities in the *unc-31* mutants (lack of dense-core vesicle release) and *egl-3* mutants (lack of neuropeptide maturation), and did not observe decreased GABA_AR recruitment in those mutants. These data suggest that neither dense-core vesicle release nor neuropeptide secretion is required for transsynaptic GABA_AR recruitment, and support the conclusion that UNC-43 recruits GABA_ARs independent of dense-core vesicle release or neuropeptide secretion.

(3) We found GABA_ARs recruitment is not regulated by the BK channel SLO-1

Prompted by the reviewer's suggestion, we also examined whether UNC-43/CaMKII recruits GABA_ARs through the BK channel SLO-1, as previous research showed that *unc-43* gain-of-function mutation inhibits ePSC through SLO-1 (PMID: 17898212). However, we observed no alteration of GABA_ARs puncta fluorescence intensities and densities in the *slo-1* loss-of-function mutants, suggesting SLO-1 is not involved in recruiting postsynaptic GABA_ARs and supporting that UNC-43 recruits GABA_ARs independent of SLO-1.

In summary, these data suggest that the GABA_ARs recruitment defect in the *unc-43* mutants does not result from an abnormal synaptic vesicle or dense-core vesicle release, and is independent of SLO-1. Our new data separates UNC-43's regulatory roles in MADD-4B and NRX-1 α from its known presynaptic functions/targets including synaptic vesicle release, dense-core vesicle release, and SLO-1. We hope our newly added data addresses the reviewer's concern.

We added the data in Supplementary Figure 6, and revised the manuscript on pages 9-10, which reads as follows:

"We also ruled out the possibility that the decreased postsynaptic GABA_ARs in the *unc-43* mutants result from diminished synaptic GABA transmission or neuropeptide release, as both a previous report and our data have demonstrated that the synaptic abundance of GABA_AR was not decreased in the *unc-25* mutants (which lack GABA biogenesis), *snb-1* mutants, *unc-13* mutants (which lack synaptic vesicle release), *unc-31* mutants (which lack dense-core vesicle release), and *egl-3* (which lack neuropeptide maturation) (Supplementary Fig. 6A-C)⁶⁶. The increase of GABA_ARs puncta fluorescence in the *unc-25* mutants may be caused by synaptic homeostasis. Further, GABA_ARs recruitment is not modulated by the BK channel SLO-1⁴⁹ (Supplementary Fig. 6A)."

A**B****C**
Supplementary Figure 6. Lack of synaptic vesicle or dense-core vesicle release does not cause a decrease in GABA_AR abundance at synapses. TagRFP-UNC-49 puncta fluorescence intensities in *unc-25* (A), *slo-1*, *unc-13*, *snb-1*(B), *unc-31*, and *egl-3* (C) mutants. Representative images (scale bar 10 μ m) and mean puncta intensity are shown. Pseudo-color: Red Hot. Ns represent the number of animals tested. ***, $p < 0.001$. Unpaired Student's t-test for A, one-way ANOVA with post-hoc Bonferroni's multiple comparison test for B and C.

B) The authors argue for an exclusive presynaptic role for CaMKII. The genetic rescue experiments do not fully support this conclusion, nor does an earlier electrophysiological study in *C. elegans*.

Response:

Please see “Response to specific comments 2”.

C) The plasticity experiments are very preliminary and difficult to interpret.

Response:

Please see “Response to specific comments 5”.

No discussion of relevant work studying the inhibitory LTP in mammals and the role of postsynaptic CaMKII.

Response:

We have revised the discussion on page 16, lines 11-21, which reads as follows:

“Most previous studies of CaMKII have focused on its facilitation of AMPAR recruitment to enhance synaptic strength at postsynapses^{29,32}. At inhibitory synapses, CaMKII is reported to phosphorylate GABA_ARs and regulates their surface delivery³⁶⁻⁴⁰. In hippocampal neurons, the moderate N-methyl-D-aspartate receptor (NMDAR)-activating stimuli cause CaMKII selectively translocating to inhibitory synapses to phosphorylate GABA_AR β 3S383 and recruit scaffold protein gephyrin, which promotes GABA_ARs accumulation and immobilization at synapses^{2,44,70}. Besides, an acute increase in neuronal activity of cultured hippocampal neurons also promotes the phosphorylation of β 3S383 by CaMKII³⁸. Further, in cerebellar somatodendritic basket cell synapses, CaMKII is required for the rebound potentiation (RP) by phosphorylation of GABA_ARs receptor⁴³.”

Specific Comments:

1. The authors do not provide the necessary background to motivate the study of presynaptic CaMKII with respect to the regulation and plasticity of postsynaptic GABA-A receptors, thus the choice to study CaMKII appears somewhat arbitrary. A rich vertebrate literature argues that postsynaptic CaMKII regulates postsynaptic receptors and inhibitory LTP (e.g., Petrini et al., Nature Comm, 2014, which provides a detailed mechanistic analysis; Houston et al., J. Physiol, 2009). The authors introduce their study by speculating that transsynaptic recruitment of GABA-A receptors might occur, and if so, CaMKII might have a role in this process. The study of CaMKII appears motivated solely by the importance of this protein for mammalian LTP. The authors do not adequately discuss the vertebrate literature in the introduction nor do they place their findings in the context of a larger literature in the discussion.

Response:

We appreciate the reviewer for raising this important concern. **According to the reviewer’s suggestion, we have revised the Introduction on page 4, lines 14-20, and the discussion on page 16, lines 11-21, which reads as follows:**

“CaMKII is required for the induction of rebound potentiation in Purkinje neurons⁴¹⁻⁴³ and is also required for the moderate N-methyl-D-aspartate receptor (NMDAR)-activating stimuli-induced long-term potentiation of inhibition (iLTP) in hippocampal neurons^{2,44}. Besides that, it has been known for a long time that CaMKII is expressed and functions on the presynaptic side^{34,45-50}. However, whether the presynaptic CaMKII could transsynaptically recruit postsynaptic receptors and be involved in inhibitory synaptic plasticity remain elusive.”

“Most previous studies of CaMKII have focused on its facilitation of AMPAR recruitment to enhance synaptic strength at postsynapses^{29,32}. At inhibitory synapses, CaMKII is reported to phosphorylate GABA_ARs and regulates their surface delivery³⁶⁻⁴⁰. In hippocampal neurons, the moderate N-methyl-D-aspartate receptor (NMDAR)-activating stimuli cause CaMKII selectively translocating to inhibitory synapses to phosphorylate GABA_AR β 3S383 and recruit scaffold protein gephyrin, which promotes GABA_ARs accumulation and immobilization at synapses^{2,69,70}. Besides, an acute increase in neuronal activity of cultured hippocampal neurons also promotes the phosphorylation of β 3S383 by CaMKII³⁸. Further, in cerebellar somatodendritic basket cell synapses, CaMKII is required for the rebound potentiation (RP) by phosphorylation of GABA_ARs receptor⁴³.”

2. The major finding of the paper is the reduced fluorescence intensity of a tagRFP-GABA-A receptor fusion protein in CaMKII mutants (Fig. 1). The decrease is modest.

Response:

(1) We showed that there is a 35% decrease in tagRFP-GABA_ARs fluorescence intensity in the *unc-43* loss-of-function mutants.

(2) We provided mIPSC recording to support that the GABA_ARs synaptic recruitment defect in the *unc-43* mutants caused impairment of GABAergic synaptic transmission (Figure 3).

The authors argue that the defect can be rescued by selective expression of CaMKII in presynaptic GABAergic neurons. However, the rescue is only partial, calling into question the interpretation of the results. Indeed, partial rescue is found in other figures as well (e.g., Fig. 3, 4).

The issue of partial rescue is especially important since an earlier paper (Liu et al, J. Neurosci, 2007) investigated the role of CaMKII in GABA-A receptor mediated signaling and suggests that CaMKII has both presynaptic (regulation of synaptic vesicle/neurotransmitter release) and postsynaptic (surface expression of GABA-A receptors) roles. The authors argue that the incomplete rescue might be due to a requirement for multiple CaMKII isoforms. Alternatively, rescue might be dependent on the strength of the promoters. There is no independent measurement of how much CaMKII is expressed in the different cell types. These alternative models can be easily addressed with additional rescue experiments using different isoforms, including cell-specific expression of the CaMKII(gf) variant. Additionally, the authors should provide control experiments demonstrating the percent rescue using the native CaMKII promoter.

Response:

We appreciate the reviewer’s constructive suggestion. We did the following experiments to address the reviewer’s concern:

(1) We found that both UNC-43D and UNC-43G isoforms are able to partially rescue the GABA_ARs recruitment defects in the *unc-43* mutants.

According to previous research, both UNC-43D and UNC-43G have been shown to be able to mediate UNC-43's synaptic roles (PMID: 10647013; PMID: 10647014; PMID: 17898212; PMID: 23663262). Here we used the UNC-43D isoform to perform the rescuing experiments. In the revision process, we also tested the rescuing effect of UNC-43G isoform. Similar to UNC-43D, expressing UNC-43G in the GABAergic motor neurons (under *unc-25* promoter) partially rescued the GABA_ARs recruitment defects.

(2) Restoring UNC-43D expression under its own promoter fully rescues the GABA_ARs recruitment defects in the *unc-43* mutants.

According to the reviewer's suggestion, we drove UNC-43D isoform expression under its endogenous promoter in the *unc-43* mutants, and the GABA_ARs puncta fluorescence defects in the *unc-43* mutants can be fully rescued, which rules out the possibility that the partial rescue of the synaptic abundance of GABA_ARs may be caused by the involvement of multiple UNC-43 isoforms, pointing toward the possibility that UNC-43 from other cells also participate in recruiting GABA_ARs.

(3) Restoring UNC-43D expression in pan-neurons or in both GABAergic and cholinergic motor neurons fully rescues the GABA_ARs recruitment defects in the *unc-43* mutants.

Further, we restored UNC-43 (UNC-43D isoform) expression in pan-neurons (under *rab-3* promoter), in both GABAergic and cholinergic motor neurons (*unc-25* promoter+*unc-17* promoter), and in both GABAergic motor neurons and muscle cells (*unc-25* promoter+*myo-3* promoter) in the *unc-43* mutants. We found that the GABA_ARs recruitment defects in the *unc-43* mutants can be fully rescued by expressing UNC-43 in pan-neurons or in both GABAergic and cholinergic motor neurons, but not with both GABAergic motor neurons and muscle cells groups. The data suggest that both the cholinergic and GABAergic UNC-43 are involved GABA_ARs recruitment, ruling out the involvement of postsynaptic UNC-43.

(4) The cholinergic UNC-43 also participates in recruiting GABA_ARs, and it relies on the GABAergic UNC-43.

Recalling the results in Figure 1C and Figure 3A-C that restoring UNC-43 expression in the cholinergic neurons alone has no effects on rescuing both GABA_ARs recruitment defects nor mIPSC defects in *unc-43* mutants. Collectively, all of the data support the conclusion that the cholinergic UNC-43 also participates in recruiting GABA_ARs at inhibitory synapses with some unknown mechanisms, and it relies on the GABAergic UNC-43.

In summary, our data support the conclusion that the UNC-43 in the GABAergic motor neurons recruits postsynaptic GABA_ARs. Although the UNC-43 in the cholinergic motor neurons is able to additively recruit GABA_ARs, it relies on the UNC-43 in the GABAergic motor neurons. We hope our newly added data addresses the reviewer's concern. We appreciate the reviewer's constructive suggestion, which leads us to new discoveries. We hope we are able to elucidate how UNC-43 in the cholinergic neurons facilitates GABA_AR recruitment in our next paper.

Supplementary Figure 3. Rescue of GABA_ARs recruitment defect in *unc-43* mutants. The decreased TagRFP-UNC-49 puncta fluorescence intensity was rescued by transgenic expression of UNC-43 D or G isoform in GABAergic motor neuron (under *unc-25* promoter), endogenous expressing cells (under *unc-43* promoter), pan neuron (under *rab-3* promoter) both GABAergic motor neurons (under *unc-25* promoter) and cholinergic motor neurons (under *unc-17* promoter), and both in GABAergic motor neurons (under *unc-25* promoter) and body-wall muscles (under *myo-3* promoter). Representative images (left, scale bar 10 μ m) and mean puncta intensities (right) are shown. Pseudo-color: Red Hot. Ns represent the number of animals tested. Data corresponding to scatter plot labeled with different letters are significantly different ($p < 0.05$). One-way ANOVA with two-stage linear step-up procedure of Benjamini, Krieger, and Yekutieli correction for multiple comparisons.

We added the data in Supplementary Figure 3, and revised the manuscript on page 7, lines 4-14, which reads as follows:

" Since we observed a partial rescue of GABA_ARs recruitment defects in *unc-43* mutants by expressing UNC-43 in the GABAergic motor neurons, therefore, we restored UNC-43 (UNC-43D isoform) expression in pan-neurons (under *rab-3* or *unc-43* promoter), in both GABAergic and cholinergic motor neurons (under *unc-25* promoter+*unc-17* promoter), and in both GABAergic motor neurons and muscle cells (under *unc-25* promoter+*myo-3* promoter) in the *unc-43* mutants, and we found that the GABA_ARs puncta fluorescence defects in the *unc-43* mutants can be fully rescued by expressing UNC-43 in pan-neurons or in both GABAergic and cholinergic motor neurons, but not with both GABAergic motor neurons and muscle cells groups (Supplementary Fig. 3). These data suggest that the cholinergic UNC-43 is also involved in GABA_ARs recruitment with unknown mechanisms, and it depends on the GABAergic UNC-43."

3. The authors argue that CaMKII regulates MADD-4 secretion and the presynaptic delivery of the transmembrane protein NRX-1. With respect to secretion of MADD-4 secretion (Fig. 6), controls are required to address whether the observed secretion defect is due to a general defect in protein secretion by presynaptic neurons.

Response:

We appreciate the reviewer's rigorous thinking. To study whether UNC-43/CaMKII regulates general presynaptic protein secretion, we expressed a constitutive secretion GFP from the GABAergic motor neurons (under *unc-25* promoter), and measured GFP endocytosis by the scavenger cell coelomocytes. We observed no significant differences in GFP fluorescence intensities in coelomocytes between wild type and *unc-43* mutants, suggesting that UNC-43/CaMKII does not regulate general protein secretion in the GABAergic motor neurons.

We have added the data in Supplementary Figure 9, and revised the manuscript on page 13, lines 2-6, which reads as follows:

"Here we ruled out the possibility that the decreased MADD-4B-GFP fluorescence intensity in coelomocytes in *unc-43* mutants is caused by abnormal coelomocyte functions or general defect in presynaptic protein secretion, as the endocytosis of a constitutive GFP secreted from muscle cells or from the GABAergic motor neurons by coelomocytes was not affected by *unc-43* mutation (Supplementary Fig. 9)."

B

Supplementary Figure 9. The endocytosis of coelomocytes and general presynaptic protein secretion are not affected by *unc-43* mutation. (B) The constitutive secretion of GFP from GABAergic motor neuron terminals was not affected in the *unc-43* mutants. Secretion of GFP was measured by analyzing GFP fluorescence intensities in the coelomocytes. Scale bar 5 μ m. The representative images (left panel) and the mean fluorescence intensities are shown (right panel). n.s. not significant. Mann Whitney test.

With respect to surface expression of NRX-1 (Fig. 7), by eye the example images do not appear different – if anything, the overall fluorescence appears decreased in mutants compare to wild type. The interpretation of these experiments is difficult since essential controls are lacking (e.g., changing intracellular pH). Thus, the observed change in NRX-1 expression could simply reflect *unc-43*-mediated changes in intracellular pH.

Response:

(1) Here we expressed a dual-tagged NRX-1 α fusion protein by inserting pHluorin at the N-terminus and wrmScarlet at the C-terminus of NRX-1 α under the *unc-25* promoter, so the GFP fluorescent signals indicate the NRX-1 α on the surface membrane, and the wrmScarlet signals indicate the total NRX-1 α at the GABAergic motor neuron axon terminals. Here, both the representative images and quantification analyses showed that the GFP fluorescent intensities were significantly decreased in the *unc-43* mutants compared to wild type, while the wrmScarlet fluorescent intensities were not differed between the two groups, indicating that the surface delivery of NRX-1 α is decreased in the *unc-43* mutants.

(2) The strategy of using pHluorin to measure surface delivery and exocytosis is widely used in the field (see below).

1. Diverse modes of synaptic signaling, regulation, and plasticity distinguish two classes of *C. elegans* glutamatergic neurons. 2017. *Elife*
2. Autocrine BDNF–TrkB signalling within a single dendritic spine. 2016. *Nature*
3. PKA Activation Bypasses the Requirement for UNC-31 in the Docking of Dense Core Vesicles from *C. elegans*. Neurons. 2016. *Neuron*
4. The same synaptic vesicles drive active and spontaneous release. 2011. *PNAS*
5. Newly produced synaptic vesicle proteins are preferentially used in synaptic transmission. 2018. *PNAS*

(3) To rule out the possibility that the pHluorin fluorescence contains signal from the intracellular pHluorin-GABA_ARs (due to the changes of intracellular pH in the *unc-43* mutations), we acidified the extracellular environment to pH 5.5 by injecting of the pH-adjusted buffer into the body cavity of *C. elegans*. We found that all the pHluorin fluorescence was quenched upon extracellular acidification, and confirmed that the observed pHluorin fluorescence comes from the member fraction of pHluorin-GABA_ARs.

We added the data in Supplementary Figure 10, and revised the manuscript on page 13, lines 11-15, which reads as follows:

“pHluorin is a pH-sensitive version of GFP, and receptors with extracellular domain tagged pHluorin are fluorescent upon insertion in the plasma membrane to expose pHluorin in the extracellular environment (PH>6). Acidification of the extracellular environment to pH 5.5 quenched all the fluorescence in both wild type and *unc-43* mutants (Supplementary Fig. 10), suggesting that all of the fluorescent signals are from the pHluorin-NRX-1 α in the plasma membrane.”

Supplementary Figure 10. pHluorin-NRX-1 α puncta fluorescence was quenched by acidification of the extracellular environment. pHluorin and wrmScarlet dual-labeled NRX-1 α was expressed in GABAergic neurons. pHluorin and wrmScarlet puncta fluorescence in the dorsal nerve cords were measured in animals with the normal (upper panel, PH > 6) or acidified (lower panel, PH=5.5) extracellular environment. Representative images are shown. Scale bar 10 μ m.

4. An earlier paper (Liu et al., J. Neurosci., 2007)), demonstrated that GABA-A receptor-mediated currents are reduced in *unc-43*/CaMKII mutants. In contrast, no defect is found in the present manuscript. The authors argue that this is because Liu et al. use 0.2 mM GABA and this paper uses 0.5 mM GABA. Both concentrations are well above the EC50 for GABA-A receptors. The authors should resolve this important discrepancy by testing currents evoked by the application of 0.2 mM GABA.

Response:

According to the reviewer's suggestion, we recorded the 0.2 mM GABA-evoked currents in the *unc-43* mutants, and they were unaltered in *unc-43* mutants compared to the wild type.

Both the intracellular and extracellular solutions used in our study are different from those in Wang's paper, and this may account for the different observations in GABA-evoked currents in the *unc-43 mutants*. Nevertheless, the decrease in Wang's paper appears to be minor (15-20%), and may not be able to explain the bigger decrease of the GABA_ARs puncta fluorescence intensities (~35%) and mIPSC frequency and amplitude in the *unc-43 mutants*.

We added the data in Supplementary Figure 4, and revised the manuscript on page 9, lines 5-12, which reads as follows:

“To investigate whether the expression and/or surface delivery of GABA_ARs are compromised in the *unc-43* mutants, we recorded the 0.2 mM and 0.5 mM GABA-evoked currents, which were unaltered in *unc-43* mutants (Fig. 3H-I and Supplementary Fig. 4). Thus, the mIPSC amplitude defect in the *unc-43* mutants is unlikely to be caused by decreased bulk expression and surface delivery of GABA_ARs. A previous study reported a minor decrease in response to pressure-applied GABA (0.2 mM) in the *unc-43* mutant⁴⁹. This discrepancy likely arises from the different protocols of recording. These results verified that UNC-43 is required for both GABA_AR synaptic recruitment and inhibitory synaptic transmission.”

Supplementary Figure 4. The GABA-evoked currents in the *unc-43* mutants were comparable to that in wild-type animals (WT). Representative responses and mean current amplitude are shown. Ns represent the number of animals tested. n.s. not significant. Unpaired Student’s t-test.

5. In Fig. 8 the authors present evidence for activity dependent changes in GABA-A receptors (possible plasticity). This data is difficult to assess because of the lack of foundation/experimental details. By what definition is this plasticity? What are the properties of this plasticity? How does CaMKII contribute? Is the plasticity occluded by gain-of-function CaMKII?

Response:

(1) We observed the increased GABA_ARs recruitment upon GABAergic motor neuron excitation, which lasts within 2 hours.

Figure 8. Induction of activity-dependent plasticity at GABAergic synapses. (B-C) The activity-dependent plasticity at GABAergic synapses lasts within two hours. Representative images (B) and the normalized tagRFP fluorescence intensity before and after GABAergic motor neuron excitation were shown. Ns represent the number of animals tested. *** $p < 0.001$, n.s. not significant. Unpaired Student's t-test.

(2) Further, the colocalization coefficient of GABA_ARs with presynaptic marker UNC-57 is increased slightly but significantly upon GABAergic motor neuron excitation.

Figure 8. Induction of activity-dependent plasticity at GABAergic synapses. (D-E) The colocalization coefficient between GABA_ARs and GABAergic synaptic boutons was increased after excitation of GABAergic motor neurons; Pearson's correlation coefficients between the intensities of GABAergic bouton marker UNC-57 (green, labeled by UNC-57-split GFP under *unc-25* promoter) and postsynaptic GABA_ARs (red) were used to assess the localization of GABA_ARs at inhibitory synapses. Representative images (D-top, scale bar, 10 μm), corresponding line scan curve (D-bottom), and mean Pearson's correlation coefficients Ns represent the number of animals tested. *** $p < 0.001$ Unpaired Student's t-test.

(3) Both the mIPSCs frequency and amplitude were significantly increased in those animals with GABAergic motor neurons excited compared to controls without all-trans retinal (ATR), supporting that the inhibitory synaptic transmission was also increased.

Figure 8. Induction of activity-dependent plasticity at GABAergic synapses. (F-H) Endogenous inhibitory synaptic transmission was assessed by recording mIPSCs from body-wall muscles in transgenic animals expressing a channelrhodopsin variant (ChIEF) in the GABAergic motor neurons (under *unc-25* promoter) after blue-light stimulation with or without all-trans retinal (ATR). Representative mIPSC traces (F), mIPSC rates (G), and the cumulative fraction of mIPSC amplitude (H) are shown. Ns represent the number of animals tested. ** p < 0.01, *** p < 0.001, Unpaired Student's t-test for G, Kolmogorov-Smirnov test for H.

(4) The presynaptic localization of UNC-43 is increased upon GABAergic motor neuron excitation.

To study whether the increased GABA_ARs recruitment upon GABAergic motor neuron excitation potentiates inhibitory synaptic transmission, we patch-clamped body-wall muscles and recorded mIPSCs. Both the mIPSCs frequency and amplitude were significantly increased in those animals with GABAergic motor neuron excited compared to controls without all-trans retinal (ATR), supporting that the inhibitory synaptic transmission was also increased. This result indicates the existence of activity-dependent synaptic plasticity at GABAergic synapses.

Figure 9. UNC-43/CaMKII is required for activity-dependent plasticity at GABAergic synapses. (C) The GABAergic presynaptic localization of UNC-43 was increased after GABAergic motor neuron excitation by ChIEF activation. split GFP labels endogenous UNC-43 in GABAergic neurons. Representative images and mean dorsal cord puncta intensity are shown. *p < 0.05, Mann Whitney test for C.

(5) The activity-induced plasticity cannot be blocked by the *unc-43* gain-of-function mutations.

The results suggested that the increase in the presynaptic abundance of UNC-43 and constitutive activation of UNC-43 can additively promote the GABA_AR's recruitment.

Figure 9. UNC-43/CaMKII is required for activity-dependent plasticity at GABAergic synapses. (A) Presynaptic UNC-43 is required for activity-dependent GABAergic plasticity. TagRFP-UNC-49 puncta fluorescence intensity was increased after GABAergic motor neuron excitation by ChIEF activation in wild type and in the *unc-43* gain-of-function mutants, but not in the *unc-43* loss of function mutants, and this was rescued by transgenic expression of UNC-43 in GABAergic motor neurons (G. rescue). Representative images (left, scale bar 10 μ m) and mean puncta intensity (right) are shown. Pseudo-color: Red Hot, Ns represent the number of animals tested. * $p < 0.05$, ** $p < 0.01$, n.s. not significant. Kruskal-Wallis test with two-stage linear step-up procedure of Benjamini, Krieger, and Yekutieli correction for multiple comparisons.

We have added the data in Figure 8 and Figure 9A, and revised the manuscript on pages 14-15.

We would like to take this opportunity to express our gratitude for the helpful guidance from the reviewer about how to improve our study and manuscript. Many thanks.

REVIEWERS' COMMENTS

Reviewer #1 (Remarks to the Author):

The authors have and addressed my critiques carefully and thoroughly by performing additional experiments. This study advances our understanding of the molecular mechanisms of postsynaptic receptor clustering driven by presynaptic signals.

Reviewer #2 (Remarks to the Author):

I have no further comments

Reviewer #3 (Remarks to the Author):

The authors have satisfactorily addressed all my concerns/comments.

Reviewer #4 (Remarks to the Author):

The revised manuscript by Hao and co-authors addresses the role of CaMKII on signaling by postsynaptic GABA-A receptors at the *C. elegans* NMJ. Specifically, they ask whether presynaptic signaling by CaMKII regulates the clustering and function of postsynaptic GABA-A receptors. The authors use *in vivo* imaging of fluorescently tagged proteins and electrophysiology to test their model and conclude that UNC-43/CaMKII functions solely in presynaptic GABAergic motor neurons to regulate GABA-ergic signaling, specifically via regulating the secretion of MADD-4B and the surface expression of NRX-1. By this mechanism the activity of presynaptic CaMKII enhances postsynaptic GABA-A receptor localization/stabilization, and thus GABA-mediated synaptic transmission.

The authors use technically state-of-the-art approaches to investigate receptor localization and function and have gone to considerable lengths to address the concerns of the reviewers. While it is clear that there are presynaptic roles for CaMKII, shown here and earlier by others, the magnitude of the effect and the interpretation of the results, especially with respect to the previous literature, raise concerns that remain despite the conscientious efforts of the authors.

By far the biggest effect of disrupting CaMKII function is on presynaptic release of GABA. Thus, the mIPSC rate in *unc-43* mutants is only about 15% of that of wild-type controls. This effect is hard to disentangle from any presynaptic effects on clustering the receptor. The dependence of mini amplitude in *unc-43* mutants is much more modest, and as reported earlier (Liu et al, 2007) is about 75% that of control. A similar size effect is seen here.

The authors argue that GABAergic neurotransmission does not contribute to clustering, but curiously find a substantial effect in the *unc-25* mutant, which should eliminate GABA release. They attribute this to homeostatic compensation, but this is untested, and also at odds with the results in the *unc-13* and *snb-1* mutant backgrounds. In short, the data are confusing

With respect to the reduced size of the mini IPSC, the authors now argue that this is regulated in part by CaMKII in presynaptic GABAergic neurons, via regulation of MADD-4B and NRX-1, and in part in cholinergic neurons, by an unknown mechanism. The mechanism described here is only part of a complicated process.

The authors rely extensively on electrophysiological analysis to test their model. I previously noted that an earlier paper (Liu et al., *J. Neurosci.*, 2007), demonstrated that GABA-A receptor-mediated currents are reduced in *unc-43*/CaMKII mutants, indicating a dependence of GABARs on *unc-43* that cannot be explained by the presynaptic clustering model. In contrast, no defect was found in the present manuscript, which the authors argued as support for their clustering model. The authors addressed this discrepancy by using the same recording conditions as previously described

by Liu et al. However, their data is still at odds with the earlier paper: they find no defect in GABA-evoked current in *unc-43* mutants. This is quite puzzling and suggests that the experiments are somehow done very differently from the earlier study. Indeed, in the Liu paper, the evoked currents are about 300% greater than in this paper. This is a dramatic difference, and the most obvious interpretation is that the authors, probably due to some technical issues, are missing about 2/3 of the current. This is concerning. If the authors are only sampling a subset of the current, it will obviously confound any interpretation of the data.

I still find that the plasticity experiments are preliminary and difficult to interpret, and should be left to a separate paper. They do not really fit in this paper, and should be buttressed by electrophysiological analysis and additional mutant studies to rule out any effects of GABAergic neurotransmission.

In summary, the authors argue for an exclusive presynaptic role for CaMKII with respect to clustering/localization of postsynaptic GABARs. The experiments do suggest at least a partial role for UNC-43 in the GABAergic neurons, but the magnitude and relative importance of the effect is not clear, and the overall conclusions while internally consistent, do not fit well with the earlier literature that demonstrates much bigger GABA-evoked currents, and a dependence of this current on *unc-43*.

RESPONSE TO REVIEWERS' COMMENTS

Responses to Reviewer #4:

The revised manuscript by Hao and co-authors addresses the role of CaMKII on signaling by postsynaptic GABA-A receptors at the *C. elegans* NMJ. Specifically, they ask whether presynaptic signaling by CaMKII regulates the clustering and function of postsynaptic GABA-A receptors. The authors use in vivo imaging of fluorescently tagged proteins and electrophysiology to test their model and conclude that UNC-43/CaMKII functions solely in presynaptic GABAergic motor neurons to regulate GABA-ergic signaling, specifically via regulating the secretion of MADD-4B and the surface expression of NRX-1. By this mechanism the activity of presynaptic CaMKII enhances postsynaptic GABA-A receptor localization/stabilization, and thus GABA-mediated synaptic transmission.

The authors use technically state-of-the art approaches to investigate receptor localization and function and have gone to considerable lengths to address the concerns of the reviewers. While it is clear that there are presynaptic roles for CaMKII, shown here and earlier by others, the magnitude of the effect and the interpretation of the results, especially with respect to the previous literature, raise concerns that remain despite the conscientious efforts of the authors.

By far the biggest effect of disrupting CaMKII function is on presynaptic release of GABA. Thus, the mIPSC rate in *unc-43* mutants is only about 15% of that of wild-type controls. This effect is hard to disentangle from any presynaptic effects on clustering the receptor. The dependence of mini amplitude in *unc-43* mutants is much more modest, and as reported earlier (Liu et al, 2007) is about 75% that of control. A similar size effect is seen here.

Response:

We appreciate the reviewer's rigorous thoughts. The reduced mIPSC frequency could result from either reduced presynaptic release, or decreased postsynaptic receptor number, as the decrease of amplitude can lead to some of the peaks below the detection threshold that were not counted as the events of response.

To further rule out the possibility that the decreased GABA_ARs recruitment in the *unc-43* mutants results from presynaptic release defects, we analyzed GABA_ARs puncta fluorescence in the *unc-43; snb-1*, and *unc-25; unc-43* double mutants. Deletion of *unc-43* causes a reduction of GABA_AR puncta fluorescence in both the *snb-1* and *unc-25* mutants that lack GABAergic transmission, indicating UNC-43 recruits GABA_ARs independent of presynaptic GABA release.

We have added this data in Supplementary Figure 6a-b, and revised our manuscript on page 10, lines 2-4, which reads as follows:

“Further, deletion of *unc-43* causes a reduction of GABA_AR puncta fluorescence in both the *snb-1* and *unc-25* mutants (Supplementary Fig. 6a-b), indicating UNC-43 recruits GABA_AR independent of GABAergic neurotransmission.”

Supplementary Figure 6. Lack of synaptic vesicle or dense-core vesicle release does not cause a decrease in GABA_AR abundance at synapses. TagRFP-UNC-49 puncta fluorescence intensities in *unc-25* (a), *slo-1*, *unc-13*, *snb-1* (b) mutants. Representative images (left, scale bar 10 μ m, Pseudo-color: Red Hot) and mean puncta intensities \pm SEM (right, Ns represent the number of animals tested) are shown. Unpaired Student's t-test for A, one-way ANOVA with post-hoc Bonferroni's multiple comparison test for B and C. * $p < 0.05$, ** $p < 0.01$, *** $p < 0.001$, n.s. not significant. For a-b, source data are provided as a Source Data file.

The authors argue that GABAergic neurotransmission does not contribute to clustering, but curiously find a substantial effect in the *unc-25* mutant, which should eliminate GABA release. They attribute this to homeostatic compensation, but this is untested, and also at odds with the results in the *unc-13* and *snb-1* mutant backgrounds. In short, the data are confusing

Response:

In the *unc-25* mutants, only the GABA release is eliminated, which alters the excitatory and inhibitory balance at the neuromuscular junctions in *C. elegans* and may induce homeostasis. While in the *unc-13* or *snb-1* mutant, both the GABA and acetylcholine releases are compromised. This could explain the differences between *unc-25*, *unc-13*, and *snb-1* mutants. We hope we are able to elucidate whether the increased GABA_AR recruitment in the *unc-25* mutants is caused by homeostatic compensation in our next paper.

With respect to the reduced size of the mini IPSC, the authors now argue that this is regulated in part by CaMKII in presynaptic GABAergic neurons, via regulation of MADD-4B and NRX-1, and in part in cholinergic neurons, by an unknown mechanism. The mechanism described here is only part of a complicated process.

Response: Although the UNC-43 in the cholinergic motor neurons is able to additively recruit GABA_ARs, it relies on the UNC-43 in the GABAergic motor neurons. We appreciate the reviewer's constructive suggestion, which leads us to new discoveries. We hope we are able to elucidate how UNC-43 in the cholinergic neurons facilitates GABA_AR recruitment in our next paper.

The authors rely extensively on electrophysiological analysis to test their model. I previously noted that an earlier paper (Liu et al., *J. Neurosci.*, 2007), demonstrated that GABA-A receptor-mediated currents are reduced in *unc-43*/CaMKII mutants, indicating a dependence of GABA_ARs on *unc-43* that cannot be explained by the presynaptic clustering model. In contrast, no defect was found in the present manuscript, which the authors argued as support for their clustering model. The authors addressed this discrepancy by using the same recording conditions as previously described by Liu et al. However, their data is still at odds with the earlier paper: they find no defect in GABA-evoked current in *unc-43* mutants. This is quite puzzling and suggests that the experiments are somehow done very differently from the earlier study. Indeed, in the Liu paper, the evoked currents are about 300% greater than in this paper. This is a dramatic difference, and the most obvious interpretation is that the authors, probably due to some technical issues, are missing about 2/3 of the current. This is concerning. If the authors are only sampling a subset of the current, it will obviously confound any interpretation of the data.

Response: We appreciate the reviewer for bringing up this discrepancy. Our 0.2 mM GABA-evoked current in the wild type is around 1.21 nA, while the current from the mentioned paper (Liu *et al. J. Neurosci.*, 2007) is around 3.5 nA. We searched the literature on the GABA-evoked currents at neuromuscular junctions in *C. elegans*, and most of the papers showed that GABA-evoked currents range from 0.7 nA to 1.7 nA (see below), which is consistent with our results. In this case, we believe our recording meets the standard in the field. The data can reflect physiological conditions and support our conclusions.

Below I listed the recordings of the GABA-evoked currents by different labs:

(1) 0.5 mM GABA-evoked currents in wild type animals are around 0.7 nA from (Kang Shen lab, *Neuron*, 2015, PMID: 26028574)

Redacted

(2) 0.1 mM GABA-evoked currents in wild type animals are around 0.8 nA from (Jean-Louis Bessereau lab, *Neuron*, 2015, PMID: 26028575)

(3) 0.1 mM GABA-evoked currents in wild type animals are around 1.7 nA from (Jean-Louis Bessereau lab, *EMBO J*, 2007, PMID: 17853888)

(4) 0.1 mM GABA-evoked currents in wild type animals are around 1.7 nA from (Erik Jorgensen lab, *Nat. Neurosci.*, 1999, PMID: 10461217)

(5) 0.1 mM GABA-evoked currents in wild type animals are around 1.3 nA from (Andres Villu Maricq lab, *Neuron*, 2005, PMID: 15944127)

Redacted

I still find that the plasticity experiments are preliminary and difficult to interpret, and should be left to a separate paper. They do not really fit in this paper, and should be buttressed by electrophysiological analysis and additional mutant studies to rule out any effects of GABAergic neurotransmission.

Response: Thanks for the reviewer's suggestion. We have added the electrophysiological analysis in Figure 8f-g. Further, our newly added data (Supplementary Figure 6a-b) demonstrated that UNC-43 recruits GABA_ARs independent of GABAergic neurotransmission

In summary, the authors argue for an exclusive presynaptic role for CaMKII with respect to clustering/localization of postsynaptic GABA_ARs. The experiments do suggest at least a partial role for UNC-43 in the GABAergic neurons, but the magnitude and relative importance of the effect is not clear, and the overall conclusions while internally consistent, do not fit well with the earlier literature that demonstrates much bigger GABA-evoked currents, and a dependence of this current on *unc-43*.

Response: The discrepancy in the size of GABA-evoked currents could result from different intracellular and extracellular solutions and different protocols used for recording. However, our current size is consistent with the published data from different labs (as shown in our above responses). Our data showed that GABA-evoked currents were unaltered in *unc-43* mutants.

We appreciate the reviewer's rigorous thoughts to improve our study. We do hope our new data and explanations are able to address the reviewer's concerns.